# The non-canonical mitochondrial inner membrane presequence translocase of trypanosomatids contains two essential rhomboid-like proteins

Anke Harsman[1], Silke Oeljeklaus[2], Christoph Wenger[1], Jonathan L. Huot[1], Bettina Warscheid[2,3] & André Schneider[1]

Mitochondrial protein import is essential for all eukaryotes. Here we show that the early diverging eukaryote *Trypanosoma brucei* has a non-canonical inner membrane (IM) protein translocation machinery. Besides TbTim17, the single member of the Tim17/22/23 family in trypanosomes, the presequence translocase contains nine subunits that co-purify in reciprocal immunoprecipitations and with a presequence-containing substrate that is trapped in the translocation channel. Two of the newly discovered subunits are rhomboid-like proteins, which are essential for growth and mitochondrial protein import. Rhomboid-like proteins were proposed to form the protein translocation pore of the ER-associated degradation system, suggesting that they may contribute to pore formation in the presequence translocase of *T. brucei*. Pulldown of import-arrested mitochondrial carrier protein shows that the carrier translocase shares eight subunits with the presequence translocase. This indicates that *T. brucei* may have a single IM translocase that with compositional variations mediates import of presequence-containing and carrier proteins.

[1] Department of Chemistry and Biochemistry, University of Bern, Freiestrasse 3, Bern CH-3012, Switzerland. [2] Department of Biochemistry and Functional Proteomics, Institute of Biology II, Faculty of Biology, University of Freiburg, Schänzlestraße 18, Freiburg 79104, Germany. [3] BIOSS Centre for Biological Signalling Studies, University of Freiburg, Schänzlestraße 18, Freiburg 79104, Germany. Correspondence and requests for materials should be addressed to B.W. (email: bettina.warscheid@biologie.uni-freiburg.de) or to A.S. (email: andre.schneider@dcb.unibe.ch).

Emergence of a mitochondrial protein import system was a crucial factor allowing the conversion of the endosymbiotic ancestor of the mitochondrion into a true nucleus-controlled organelle. Thus, protein import is essential for the biogenesis of present-day mitochondria in all eukaryotes, since the vast majority of the more than 1,000 mitochondrial proteins are synthesized on cytosolic ribosomes[1].

The best studied mitochondrial protein import system is that of yeast. Up to 70% of all mitochondrial proteins carry an N-terminal mitochondrial targeting signal (MTS) directing them to the most commonly used route for import, the presequence pathway[2]. The proteins are first imported into mitochondria by the translocase of the outer mitochondrial membrane (TOM), which is the general entry gate for essentially all mitochondrial proteins. The TOM complex consists of the pore-forming β-barrel protein Tom40, the three import receptors Tom20, Tom22 and Tom70 and three small proteins Tom5, Tom6 and Tom7 (refs 3,4). From the TOM complex presequence carrying proteins are handed over to the presequence translocase of the inner mitochondrial membrane, the TIM23 complex[3]. Tim23 in tight association with Tim17 forms the translocation pore of this machinery[5]. The two proteins are homologous and members of the Tim17/22/23 protein family. Together with Tim50 and Mgr2 they form the membrane-embedded core of the pathway, which is modulated by the non-essential accessory components Tim21 and Pam17 (refs 6–8). The presequence translocase-associated motor (PAM) in cooperation with the mitochondrial inner membrane (IM) potential drives translocation of substrate proteins or domains thereof into the matrix using mitochondrial heat shock protein 70 (mtHsp70)-mediated ATP hydrolysis[9,10]. The essential proteins Tim44, Pam18 and Pam16 promote coupling of mtHsp70 to TIM23 and regulate its activity[3,8,11]. Import of proteins across the IM requires the cooperation of the TOM and the TIM23 complexes. This allows purification of a TOM-TIM-preprotein complex when tightly folded domains in the C-terminus of the import substrate prevent complete translocation across the outer mitochondrial membrane[12,13].

Mitochondrial proteins that are synthesized without a cleavable presequence also enter mitochondria via the TOM complex but are subsequently sorted to distinct import pathways by internal as yet ill-defined signals[1]. Hydrophobic IM proteins with multiple membrane-spanning domains, such as mitochondrial carrier proteins (MCPs), are imported by TIM22, the carrier translocase of the IM[14,15]. Tim22, a member of the Tim17/22/23 protein family, forms the translocation channel of this complex[16,17]. Furthermore, four accessory components, Tim54, Tim18, Sdh3 and Tim12, were shown to support MCP import in yeast[15,18]. To prevent premature folding of TIM22 substrates during transfer across the intermembrane space (IMS), the hydrophobic proteins are protected by a family of small chaperones, called tiny Tims[19,20].

The parasitic protozoon *Trypanosoma brucei* is only remotely related to yeast and mammals. It is one of the earliest diverging eukaryotes and has a fully functional mitochondrion capable of oxidative phosphorylation[21]. Its mitochondrial proteome and that of yeast are of similar size, indicating that it must import a similar number of proteins[22,23]. However, even though the general import pathways seem to be conserved[24,25], its nuclear genome encodes only few homologues of subunits of the mitochondrial protein translocases of yeast and mammals[26,27]. The archaic translocase of the outer membrane (ATOM), the trypanosomal analogue of the yeast TOM complex, consists of six subunits. Only two of them, ATOM40 and ATOM14, show a weak homology to yeast Tom40 and Tom22, respectively[28]. The other four subunits are unique to trypanosomatids and thus evolved independently of any TOM complex subunits of other eukaryotic lineages[28,29].

The situation is even more extreme for the trypanosomal TIM complex. The inner mitochondrial membrane of *T. brucei* harbours only a single member of the Tim17/22/23 family of proteins that likely is homologous to yeast Tim17 (refs 30,31). The *T. brucei* genome encodes putative orthologues of Tim50, Tim44 and Pam18. However, they either lack functional confirmation or, as in the case of TbTim50, a direct role in import is questionable since their ablation causes pleiotropic effects[27,32]. The tandem-affinity-purified TbTim17 complex was found to contain three novel trypanosome-specific proteins, TbTim47, TbTim54 and TbTim62, whose ablation appears to affect mitochondrial protein import[33]. One of these proteins, TbTim62, was recently shown to be essential for assembly and stability of the TIM complex[34].

By a set of complementary immunoprecipitations (IPs) and the purification of the protein complex containing a stalled translocation intermediate we now have elucidated the composition of the trypanosomal TIM translocase, which remarkably contains two essential proteins of the rhomboid family. In line with the presence of a single Tim17/22/23 homologue, the global analysis of the effects of TbTim17 knockdown revealed similar impact on the presequence as well as the MCP import pathway.

## Results

**TbTim17-RNAi affects presequence-containing and MCPs.** TbTim17 was proposed to be involved in the import of mitochondrial presequence-containing proteins as well as carrier proteins[30]. It has previously been shown that in *T. brucei*, as might be expected, inhibition of import leads to a decline in the abundance of mitochondrial proteins[28]. To study the function of TbTim17, we performed a quantitative proteomic analysis of the steady-state levels of mitochondrial proteins in a TbTim17-RNAi cell line using stable isotope labelling by amino acids in cell culture (SILAC) combined with high-resolution mass spectrometry (MS). Uninduced and induced TbTim17-RNAi cells were grown in medium containing different stable isotope-labelled forms of arginine and lysine. Three days after induction of RNAi equal cell numbers of uninduced and induced cultures were mixed and mitochondria-enriched fractions were prepared for further analysis by quantitative MS. At this time point, the induced cells did not show a growth phenotype yet (Fig. 1a). The Volcano plot shown in Fig. 1b illustrates that 208 out of the 2,048 proteins detected in triplicate experiments showed more than 1.5-fold decreased steady-state levels 3 days after induction of RNAi against TbTim17 (see also Supplementary Data 1). The observed reduction in the abundance of mitochondrial protein is likely mainly due to inhibition of mitochondrial protein import, although increased degradation of organellar proteins that lack stoichiometric amounts of cognate binding partners may contribute to it.

We have previously determined the proteome of the outer mitochondrial membrane of *T. brucei*[22]. Moreover, the same study also defined a cluster of inner mitochondrial membrane proteins. Interestingly, while only 15% of the outer membrane (OM) proteins detected in our data set are more than 1.5-fold downregulated upon Tim17 ablation (Fig. 1c), this number rises to 52% for all detected proteins from the IM cluster and to 64% for all detected proteins that have a predicted N-terminal MTS[35]. Figure 1d furthermore shows that the mean abundance levels of seven out of the eight detected MCPs are reduced after the ablation of TbTim17.

In summary, this global analysis shows that TbTIM17 is required for import of IM proteins, regardless of whether they carry an MTS or are MCPs.

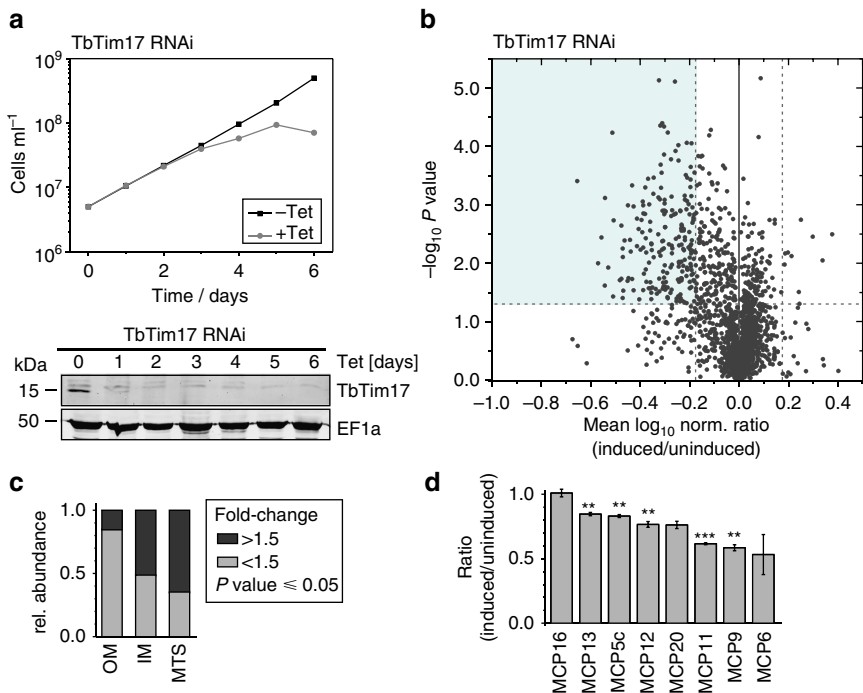

**Figure 1 | TbTim17-RNAi affects presequence-containing and carrier proteins.** (**a**) Growth of the uninduced and induced TbTim17-RNAi cell line in SILAC medium. (lower) Immunoblot demonstrating the ablation of TbTim17 during TbTim17-RNAi when assayed in SILAC medium. EF1a is shown as a loading control. (**b**) The TbTim17-RNAi cell line was subjected to SILAC-based quantitative MS comparing protein abundances in induced versus uninduced cells. For proteins quantified in at least two of three independent biological replicates, the mean $\log_{10}$ of normalized (norm.) ratios (induced/uninduced) was plotted against the $-\log_{10} P$ value (two-sided $t$-test). A $t$-test significance level of 0.05 is indicated by the dashed horizontal line. Vertical dashed lines mark a fold-change in protein abundance of $\pm 1.5$. For a complete list of proteins, see Supplementary Data 1. (**c**) Relative (rel.) abundances of all proteins from the OM proteome, the preliminary cluster of IM proteins and predicted MTS-containing proteins that were detected by quantitative MS in TbTim17 RNAi experiments. (**d**) Individual abundance ratios (induced/uninduced) of all MCPs detected in the experiment shown in **b**. For MCP20, no $P$ value (two-sided $t$-test) was assigned since the protein was only detected in two of the three biological replicates. Scale bar, s.d.; *, $0.05 \geq P \geq 0.01$; **, $0.01 \geq P \geq 0.001$; ***, $0.001 \geq P$.

**Identification of novel trypanosomal TIM complex subunits.** To analyse the composition of the TIM translocase, we used SILAC-based quantitative affinity purification MS. More specifically, *T. brucei* cell lines allowing inducible expression of epitope-tagged putative TIM complex subunits were subjected to IP using anti-tag antibodies. Subsequently, the eluted proteins from a mixture of differentially labelled cells either expressing or lacking the tagged bait protein were analysed by quantitative MS in triplicate experiments to determine the protein abundance ratios.

SILAC-IPs using C-terminally triple myc-tagged TbTim17 as bait recovered 22 proteins that were enriched more than fivefold and an additional 19 proteins that were enriched twofold (Fig. 2a and Supplementary Data 2). Several of the detected proteins had previously been associated with mitochondrial protein import in *T. brucei*, such as the IMS chaperones Tim9, Tim10, Tim8-13 (ref. 36) and components of the ATOM complex, ATOM40, ATOM46 and ATOM14 (refs 28,37). Moreover, TbTim62 and acetyl-CoA-dehydrogenase (ACAD) were already identified as interactors of TbTim17 by tandem affinity purification[33]. However, other previously proposed interaction partners of TbTim17 were not identified in our analysis[32,33]. In addition, several MCPs were also co-purified with TbTim17 (Supplementary Data 2).

To identify the true subunits of the TIM translocase, we performed reverse SILAC-IPs using two uncharacterized TbTim17 interactors as baits. For this, we selected TbTim42, which has no similarity to any protein import component, and a protein with a twin CX$_3$C motif characteristic for tiny Tim-like

IMS chaperones, termed TbTim13. Both proteins could efficiently be co-purified with TbTim17 (Fig. 2a).

The three sets of SILAC-IPs exhibited an overlap of nine proteins (including two of the bait proteins) that were at least twofold enriched (Fig. 2b and Supplementary Fig. 1a). The third bait protein TbTim17 was not recovered in the SILAC-IP of TbTim42 but it was detected by immunoblots in TbTim42 pulldowns increasing the overlap to 10 proteins that interact with all three baits (Supplementary Fig. 1b).

Except for Tim10 and TbTim17 itself, all of them are components of the previously defined cluster of IM proteins[22]. Six have predicted transmembrane domains (TMD) and three contain a predicted MTS (Fig. 2c). The remaining four proteins carry twin CX$_3$C motifs typical for tiny Tim chaperones. Reciprocal BLAST analysis and secondary structure similarities detected by HHPred[38] indicate that two of the proteins, termed TimRhom I and TimRhom II, belong to the rhomboid family of proteins. This protein family consists of not only intramembrane proteases that are conserved in all domains of life[39,40] but also includes a growing group of rhomboid-like proteins that are proteolytically inactive[41]. TimRhom I and TimRhom II belong to the latter group, since the Ser-His catalytic dyad is not conserved (Supplementary Fig. 2). The presence of rhomboid-like proteins in the TIM complex of *T. brucei* was unexpected. Thus, the interaction between TbTim17 and TimRhom I was confirmed by a pulldown with a TimRhom I-specific antiserum. (Supplementary Fig. 1c).

We focused our further analysis on TbTim42, TimRhom I and TimRhom II.

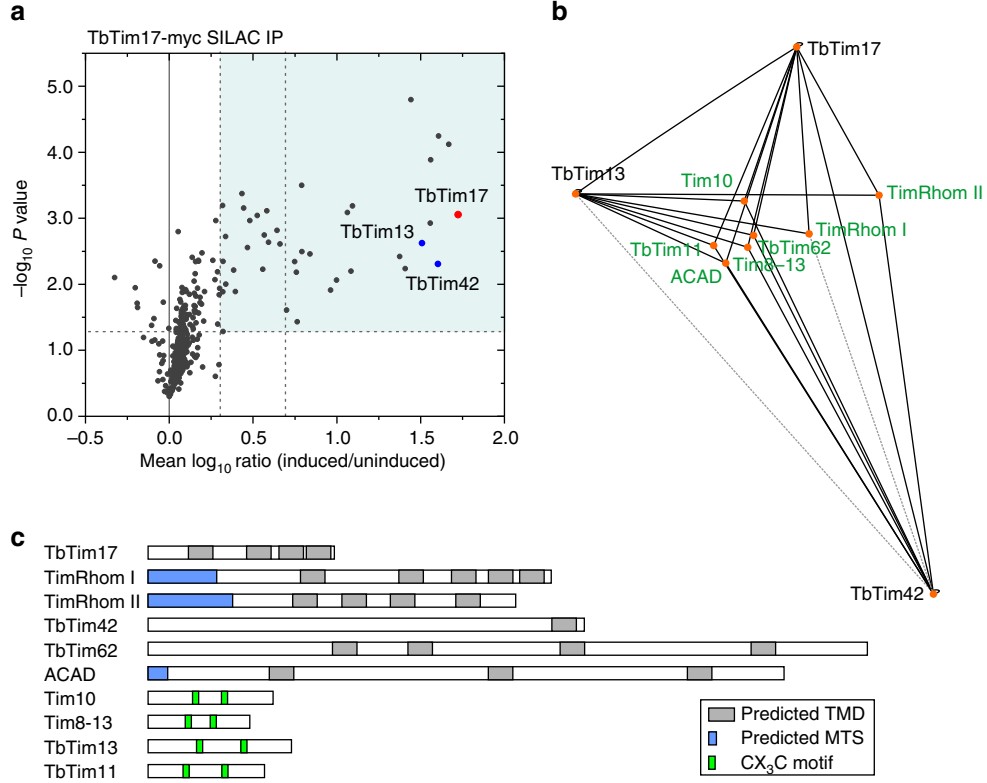

**Figure 2 | Identification of novel TIM complex subunits.** (**a**) SILAC-IP of C-terminally myc-tagged TbTim17 from digitonin-solubilized mitochondrial extracts. Mean $\log_{10}$ ratios (induced/uninduced) of proteins detected by quantitative MS in at least two of three independent biological replicates are plotted against the corresponding $-\log_{10} P$ values (one-sided $t$-test). Horizontal dashed line indicates a $t$-test significance level of 0.05, while vertical dashed lines mark a two- and fivefold enrichment, respectively. The bait protein TbTim17 is marked by a red dot. Cell lines expressing HA-tagged versions of the two TbTim17 interactors, TbTim42 and TbTim13 (blue dots), were used for reciprocal SILAC-IPs. For complete lists of proteins for all three IPs, see Supplementary Data 2–4. (**b**) A network view of the overlap between the three sets of SILAC-IPs was produced using Cytoscape[66]. The connecting lines correspond to the detected enrichment factors. Depicted are the three bait proteins, which reciprocally interact as shown by SILAC-IPs or by immunoblot, as well as all proteins specifically enriched in all three CoIPs (the network including proteins interacting with two proteins only is shown in Supplementary Fig. 1). Enrichment factors of >2-fold and >5-fold are indicated by grey dashed and black solid lines, respectively. (**c**) Predicted domain structure and targeting signals of the putative TIM complex subunits present in the overlap of the three sets of SILAC-IPs. TMD were predicted using TMPred[67]. MTS were predicted by MitoProt[68]. Four proteins contain CX$_3$C motives typical for IMS proteins.

**Novel TIM subunits are membrane proteins and form a complex.** Digitonin-based cell fractionation as well as immunofluorescence (IF) microscopy verified that the three candidate proteins exclusively co-localize with mitochondrial marker proteins (Fig. 3a,b). Moreover, in an alkaline carbonate extraction of mitochondria-enriched fractions, all three proteins were recovered in the pellet and thus behaved like the integral OM proteins voltage-dependent anion channel (VDAC) and ATOM40, respectively (Fig. 3c). The soluble IMS proteins Tim9 and cytochrome C, in contrast, were released to the supernatant. Thus, TbTim42, TimRhom I and TimRhom II are integral mitochondrial membrane proteins.

To analyse whether the candidate proteins localize to high molecular weight protein complexes as TbTim17, we combined blue native–polyacrylamide gel electrophoresis (BN–PAGE) in the first dimension with SDS–polyacrylamide gel electrophoresis (SDS–PAGE) in the second dimension. The result showed that tagged TbTim17 was most highly enriched in high molecular weight complexes ranging from 700 to 1,000 kDa (Fig. 3d)[34]. All of TbTim42 and a fraction of TimRhom I were detected in similarly sized complexes (Fig. 3d). Moreover, a second complex of approximately 400 kDa seems to contain TbTim17 and TimRhom I, but not TbTim42. Finally, significant amounts of TimRhom I were also found in small complexes of ∼150 kDa (Fig. 3d). TimRhom II could not be analysed as neither the tagged

version, nor the endogenous protein could be detected on immunoblots of BN-gels.

**Novel TIM subunits are essential for protein import.** *T. brucei* alternates between an insect vector and a mammalian host. While it is able to produce ATP by oxidative phosphorylation in the insect-stage, procyclic form, the bloodstream form (BSF) lacks respiratory chain components in its smaller mitochondrion and produces ATP by glycolysis[42]. However, mitochondrial protein import as well as mitochondrial translation and thus the mitochondrial genome are essential in both life cycle stages[28].

To examine the function of the novel TIM complex subunits, tetracycline-inducible RNAi cell lines were established. The top panels in Fig. 4a show that all three proteins are required for normal growth of procyclic form trypanosomes. The bottom panels of Fig. 4a demonstrate that the same is true for an engineered BSF cell line that, while still depending on mitochondrial protein import, can grow in the absence of mitochondrial DNA[43]. Thus, TbTim42, TimRhom I and TimRhom II, as expected for core components of the protein import machinery, are essential for growth under all tested conditions.

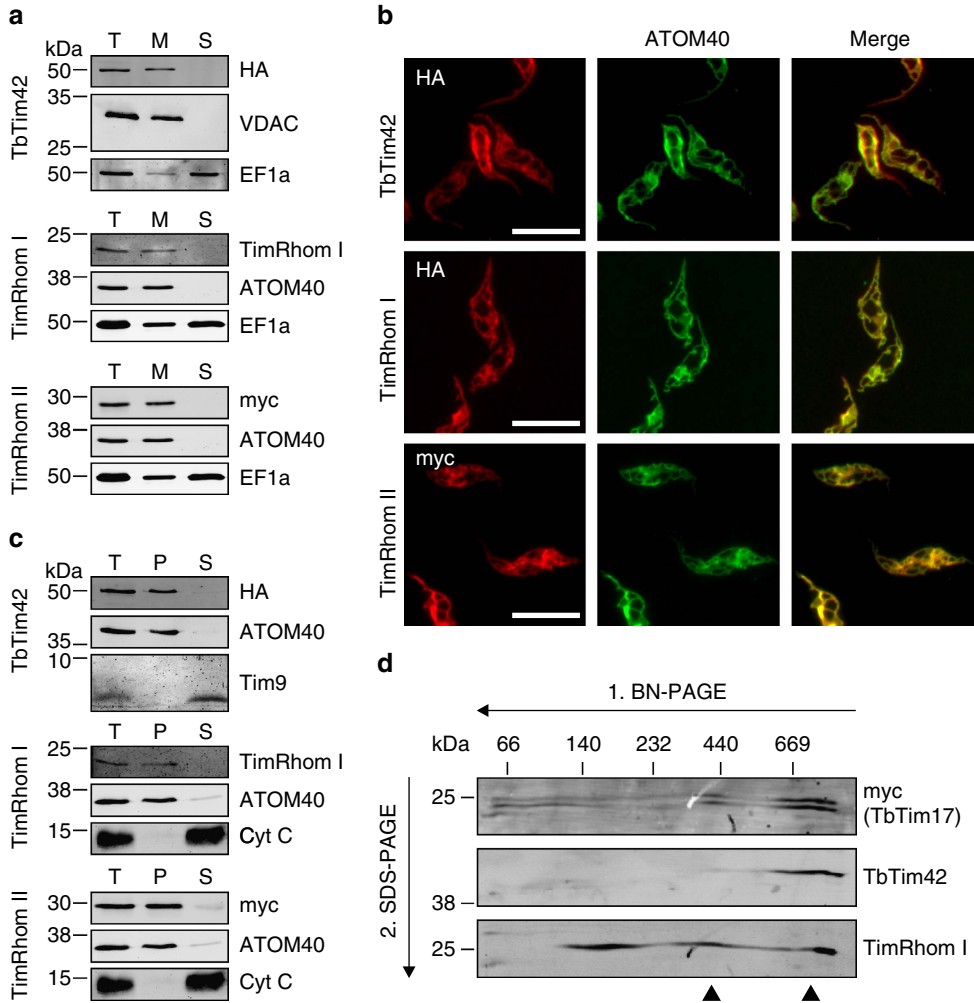

**Figure 3 | Novel TIM subunits are integral membrane proteins that form a large complex.** (**a**) Immunoblot analysis of whole-cell (T), digitonin-extracted mitochondria-enriched pellet (M) and soluble (S) fractions of cells expressing c-terminally HA-tagged TbTim42 and cells expressing c-terminally myc-tagged TimRhom II. The top and bottom panels were probed with anti-tag antibodies. The middle panel shows a redecoration of the bottom panel with a custom-made antiserum against TimRhom I, thus the same controls are depicted as in the bottom panel. ATOM40/VDAC and Elongation factor 1-alpha (EF1a) serve as markers for mitochondria and cytosol, respectively. (**b**) IF analysis of cells expressing the indicated tagged proteins. ATOM40 serves as a mitochondrial marker. Scale bar, 10 μm. (**c**) Immunoblots of the total (T), pellet (P) and supernatant (S) fractions of a carbonate extraction performed at pH 11.5. Mitochondria-enriched pellets of cells expressing c-terminally HA-tagged TbTim42 and cells expressing c-terminally myc-tagged TimRhom II were used as starting material. The top and bottom panels were probed with anti-tag antibodies. The middle panel shows a redecoration of the bottom panel with a custom-made antiserum against TimRhom I, thus the same controls are depicted as in the bottom panel. ATOM40/VDAC and cytochrome c (Cyt c)/Tim9 serve as markers for integral and peripheral membrane proteins, respectively. (**d**) Digitonin-solubilized mitochondrial protein complexes were separated in the first dimension by 6–16.5% BN–PAGE and then subjected to denaturing 14% SDS–PAGE in the second dimension. The resulting immunoblots were probed for c-terminally myc-tagged TbTim17 using an anti-myc antiserum as well as for TbTim42 and TimRhom I using custom-made antisera. Arrow heads indicate high molecular weight complexes containing TIM subunits. VDAC, voltage-dependent anion channel.

To directly demonstrate that the three proteins are involved in protein import, we analysed the steady-state levels of mitochondrial proteins in whole-cell extracts of the corresponding RNAi cell lines in insect-stage *T. brucei*. Accumulation of the unprocessed precursor of cytochrome c oxidase subunit 4 (CoxIV) in the cytosol is an established hallmark of mitochondrial protein import defects upon *in vivo* ablation of protein import factors. Under the same conditions, many other mislocalized mitochondrial precursors are rapidly degraded[28]. Indeed, accumulation of the CoxIV precursor could be observed in all three cell lines after 2 days of induction (Fig. 4b), which coincides with the onset of the growth arrest. The Mitotracker staining in Supplementary Fig. 3 shows that in all three cell lines the membrane potential was still intact at this time point. We furthermore probed the same immunoblots for TbTim17. The

middle panel on the left in Fig. 4b shows that ablation of TbTim42 destabilizes TbTim17 even before the growth arrest, suggesting a close functional interaction between the two proteins. In the induced TimRhom I and TimRhom II RNAi cell lines, however, the level of TbTim17 is stable or only slightly reduced indicating that the two rhomboid-like proteins are directly required for mitochondrial import of presequence-containing proteins even in the presence of TbTim17.

**TIM subunits present in the active presequence translocase.** Chimeric MTS-containing precursor proteins that are fused to dihydrofolate reductase (DHFR) were essential for the initial characterization of Tom40 (refs 44) as well as for the identification of its functional analogue ATOM40 in trypanosomes[37]. The reason

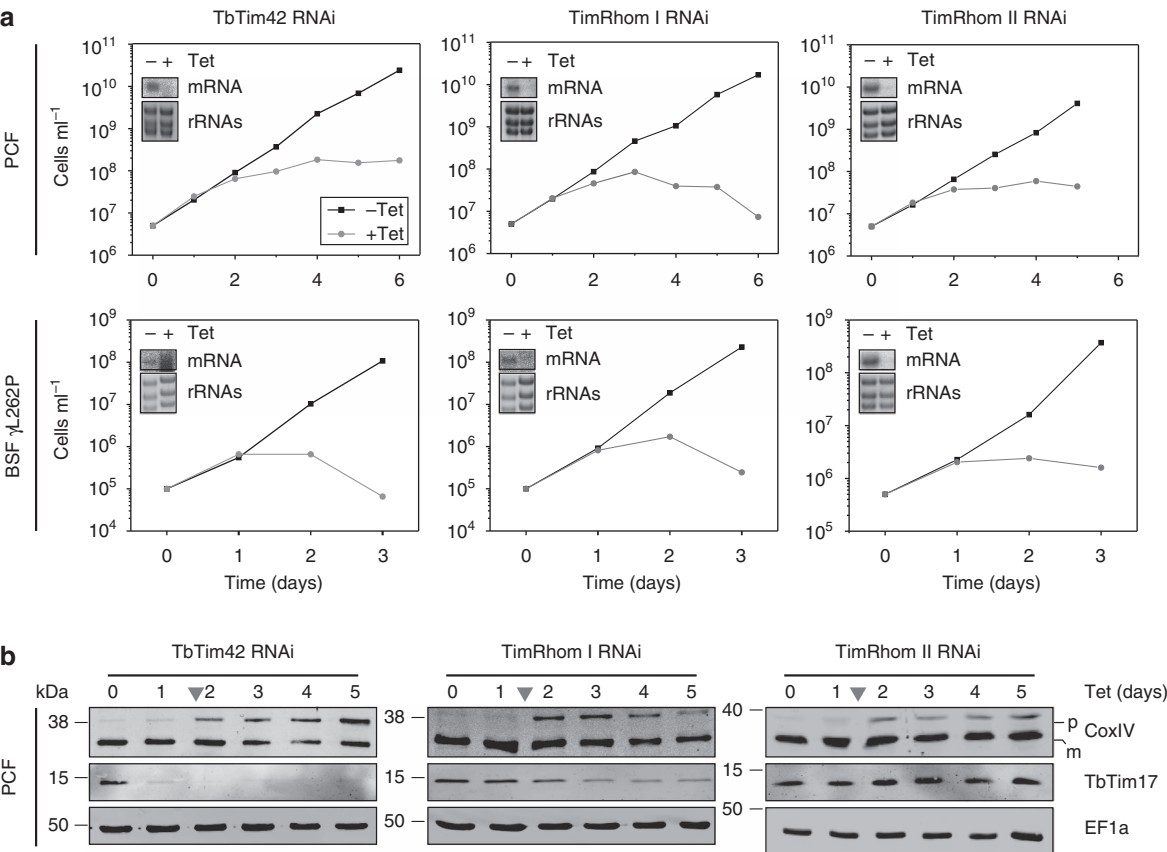

**Figure 4 | Novel TIM complex subunits are essential for protein import. (a)** Growth curves of uninduced ( − Tet) and induced ( + Tet) PCF and BSF RNAi cell lines ablating TbTim42, TimRhom I or TimRhom II. Because of the L262P mutation in the nucleus-encoded γ-subunit of the ATPase, utilized BSF cell lines are able to grow in the absence of kDNA[43]. Northern blots of total RNA extracts from uninduced ( − Tet) or 2 days induced ( + Tet) cells probed for the corresponding mRNAs are shown in the respective insets. Ethidiumbromide-stained rRNAs serve as loading controls. **(b)** Immunoblots depicting steady-state levels of CoxIV and TbTim17 in whole-cell extracts of the indicated RNAi cell lines. Cytosolic EF1a serves as a loading control. Time of RNAi induction by tetracycline is indicated at the top. Arrow heads indicate the onset of the growth phenotype. The migration positions of precursor (p) and mature forms (m) of CoxIV are marked. PCF, procyclic form.

is that the folate analogues methotrexate (MTX) or aminopterin (AMT) bind to DHFR with very high affinity and stabilize the protein in its folded form. Since mitochondrial protein import requires unfolding of the transported substrate addition of MTX/AMT will not affect membrane translocation of the N-terminal part of the chimeric protein, but it will block import of the irreversibly folded DHFR moiety[12,45]. Provided that the N-terminal part of the chimeric protein is at least 50 amino acids long, an import intermediate is formed that simultaneously blocks both the outer and the IM import channels[13,46].

Here we have used DHFR fusion proteins to identify the composition of the active presequence translocase in *T. brucei*. However instead of using isolated mitochondria we adapted the system, so that the import intermediate can be formed *in vivo*. To that end a transgenic cell line was produced that allows inducible expression of a chimeric substrate composed of the presequence (1–14 aa) and part of the mature form (15–160 aa) of the trypanosomal matrix protein dihydrolipoyl dehydrogenase (LDH), followed by DHFR carrying a C-terminal triple HA-tag[47] (Fig. 5a).

BN–PAGE of mitochondrial protein complexes from this cell line revealed that upon addition of AMT, the LDH-DHFR fusion protein shifts from the position corresponding to the monomeric form ( ∼ 50 kDa) to a high molecular weight complex. Similarly, ATOM40 is quantitatively shifted from the normal ∼ 700 kDa ATOM complex[28] to the same high molecular weight complex

that is observed for LDH-DHFR (Fig. 5b). Thus, addition of AMT induces the formation of a supercomplex, which as shown below, consists of the import substrate, the ATOM and the IM presequence translocase. As a consequence, further protein import is blocked as evidenced by the cytosolic localization of a fraction of the LDH-DHFR fusion protein (Supplementary Fig. 4a) and the accumulation of the CoxIV precursor that is observed in the presence of AMT (Supplementary Fig. 4b). The C-terminal triple HA-tag allows affinity purification of the LDH-DHFR import substrate. This can either be done in the absence of AMT when it is localized in the matrix or in the presence of AMT when a fraction of it is arrested in the import channels. The result of such an experiment is shown in the immunoblots in Fig. 5c. It demonstrates that ATOM40, TbTim17 and the newly characterized TIM subunit TimRhom I, but not the respiratory protein CoxIV, specifically co-purify with the arrested LDH-DHFR in the presence of AMT.

To identify the full complement of proteins that are specifically associated with the AMT-dependent protein translocation intermediate, we used SILAC-IP as described above (Fig. 5d). Untreated or AMT-treated cell cultures, both expressing the LDH-DHFR substrate, were subjected to SILAC-IP and analysed by quantitative MS to reveal the proteins that are specifically associated with the stalled LDH-DHFR import substrate. In contrast to the SILAC-IPs shown in Fig. 2, the approach based on the stalled intermediate detects subunits of the active import

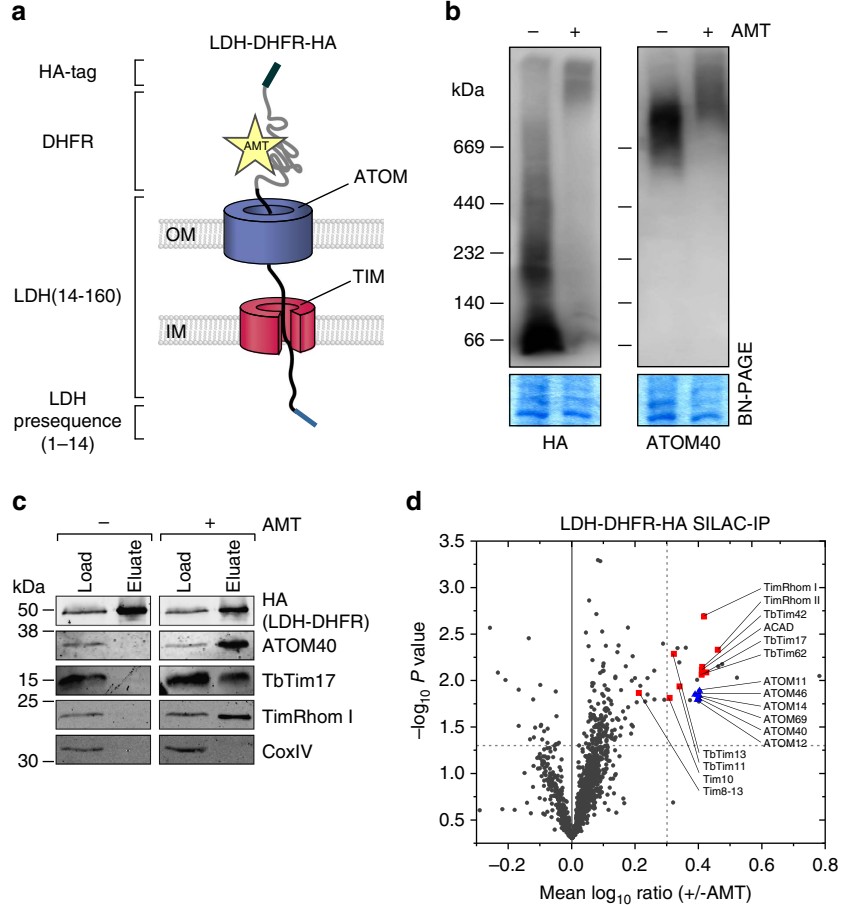

**Figure 5 | Novel TIM complex subunits are present in the active presequence translocase.** (**a**) Schematic depiction of the import intermediate induced by *in vivo* expression of the chimeric precursor protein (LDH-DHFR-HA) in the presence of AMT. (**b**) BN–PAGE of digitonin-solubilized mitochondrial protein complexes isolated from cells expressing LDH-DHFR-HA. Cells were grown in the absence and presence of AMT. (top) Immunoblots probed for LDH-DHFR-HA or for ATOM40. (bottom) Coomassie-stained gel sections serve as loading controls. (**c**) IP of LDH-DHFR-HA from digitonin-solubilized mitochondrial extracts of cells grown in the absence and presence of AMT. Five per cent of the lysate (Load) and 100% of the eluate were separated by SDS–PAGE. The resulting immunoblots were probed for ATOM40, CoxIV, as well as for two of the TIM complex subunits, TbTim17 and TimRhom I. (**d**) SILAC-IP of LDH-DHFR-HA from digitonin-solubilized mitochondrial extracts from differentially labelled cells grown in the absence and presence of AMT. Mean $\log_{10}$ ratios ( $\pm$ AMT) of mitochondrial proteins detected by quantitative MS in at least two of the three independent biological replicates are plotted against the corresponding $-\log_{10}$ *P* values (one-sided *t*-test). Dashed lines represent a mean enrichment of 2 and a *P* value of 0.05. For a complete list of proteins, see Supplementary Data 5. Blue, ATOM complex subunits; red, TIM complex subunits as defined in Fig. 2b.

machineries and does not require tagging of TIM complex components.

The obtained data set was filtered for mitochondrial proteins because the enrichment of a few non-mitochondrial proteins (Supplementary Data 5) may be related to the AMT treatment which is known to interfere with mitosis[48]. Figure 5d shows that 26 mitochondrial proteins were enriched twofold or more in the presence of AMT. As expected the group included all six subunits of the ATOM complex. Moreover, all 10 proteins defined as TIM complex subunits by the SILAC-IPs shown in Fig. 2 were detected. All but Tim8-13 were more than twofold enriched. They cluster into two groups, with membrane proteins TbTim17, TimRhom I, TimRhom II, TbTim42, ACAD and TbTim62 being more strongly enriched than the four tiny Tim-like proteins. Finally, 10 mostly uncharacterized mitochondrial proteins were also found. These might represent further components of the protein import machinery. However, neither the previously characterized Tim subunit TbTim50 (ref. 32), nor bioinformatically identified homologues of PAM components like Tim44 (refs 27) were found to be specifically enriched in our data set. This suggests that the PAM module and other peripherally

associated subunits of the TIM complex were lost during the purification procedure.

**TIM subunits present in the carrier translocase.** The proteomic analysis of the RNAi cell line in Fig. 1 suggests that TbTim17 might not only be involved in import of presequence-containing proteins but also in the biogenesis of MCPs. In order to investigate whether the subunits found in the active presequence translocase are also present in the carrier translocase, we wanted to produce an import intermediate that is stuck in the carrier import pathway. MCPs consist of three tandemly repeated structurally similar modules each of which contains two TMDs. Previous work in yeast has shown that a variant of the carboxylate carrier lacking the first module can still be imported across the OM but becomes stuck at the TIM22 complex[49]. Figure 6 shows that the same strategy to produce an import intermediate in the carrier import pathway also works in trypanosomes. C-terminally myc-tagged full-length MCP12 or a variant thereof lacking the first module were expressed in *T. brucei* (Fig. 6a). Both proteins localize to mitochondria (Fig. 6b) and according to carbonate

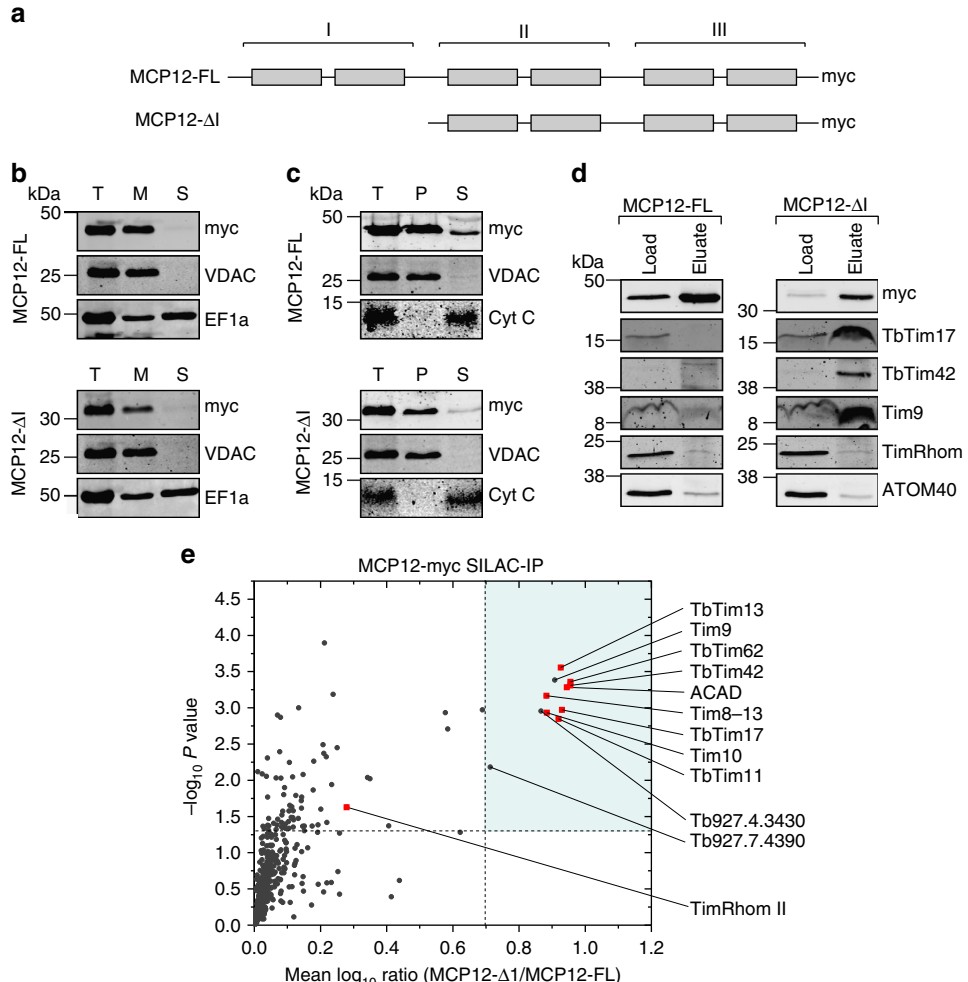

**Figure 6 | Many TIM subunits are present in the active carrier translocase. (a)** Schematic diagram of the full-length MCP12 (MCP12-FL), consisting of three structurally similar modules each containing two TMDs, and a variant thereof that lacks the first module (MCP12-ΔI). **(b)** Immunoblot analysis of whole-cell (T), digitonin-extracted mitochondria-enriched pellet (M) and soluble (S) fractions of cells expressing c-terminally myc-tagged MCP12-FL or myc-tagged MCP12-ΔI, respectively. The top panels were probed with anti-myc antibodies. The middle and the bottom panels show the staining for the mitochondrial und cytosolic markers, VDAC and EF1a, respectively. **(c)** Immunoblots of the total (T), pellet (P) and supernatant (S) fractions of a carbonate extraction (pH 11.5) from mitochondria-enriched pellets of cells expressing c-terminally myc-tagged MCP12-FL or myc-tagged MCP12-ΔI. The top panels were probed with anti-myc antibodies. VDAC and Cyt c serve as markers for integral and peripheral membrane proteins, respectively. **(d)** IPs of digitonin-solubilized mitochondrial extracts from cells expressing MCP12-FL-myc (left panel) or MCP12-ΔI-myc (right panel), respectively. Ten per cent of the lysates (Load) and 100% of the eluates were separated by SDS–PAGE. The resulting immunoblots were probed for TbTim17, TbTim42, Tim9, TimRhom I and ATOM40. **(e)** SILAC-IP of digitonin-solubilized mitochondrial extracts from differentially labelled cells expressing either full-length (MCP12-FL) or truncated variant of MCP12 (MCP12-ΔI). Mean $\log_{10}$ ratios (MCP12-ΔI /MCP12-FL) of proteins detected by quantitative MS in at least two of the three independent biological replicates are plotted against the corresponding $-\log_{10}$ P values (two-sided t-test). Dashed lines represent a mean enrichment of 5 and a P value of 0.05. For a complete list of proteins, see Supplementary Data 6. Red squares, TIM complex subunits as defined in Fig. 2b. Tb927.4.3430 shows structural similarity to tiny Tim-like chaperones and Tb927.7.4390 is annotated as putative threonine synthase.

extraction are integral membrane proteins (Fig. 6c). Immunoblots of IPs using anti-myc antibodies show that both tagged MCP12 variants are efficiently recovered in the eluate fractions (Fig. 6d). However, the TIM complex subunits TbTim17 and TbTim42 as well as the tiny TIM chaperone Tim9 are specifically co-precipitated with the truncated variant of MCP but not with the full-length protein (Fig. 6d).

To identify the full complement of putative carrier translocase subunits, we used SILAC-IP. Cell cultures expressing either the full-length or the truncated version of MCP12 were subjected to SILAC-IP and analysed by quantitative MS to reveal the proteins specifically associated with the truncated MCP12. Figure 6e shows that 11 proteins were enriched fivefold or more in IPs derived

from cells expressing the truncated MCP12. The group includes the previously described TIM subunits TbTim62, TbTim17, TbTim42, ACAD and all four previously identified tiny TIM chaperones. Interestingly, all of these proteins were also found in the active presequence translocase (Fig. 5d). Using a cutoff of fivefold only three proteins were specific for the carrier translocase but not found in the presequence translocase: the tiny TIM-like proteins TIM9 and Tb927.4.3430 as well as a putative threonine synthase. However TimRhom I and TimRhom II, which are prominent members of the active presequence translocase, were not significantly enriched in the active carrier translocase (TimRhom I was 0.8-fold and TimRhom II 1.9-fold enriched).

These results suggest that at least some of the nine TIM subunits that overlap between the two translocases do not only mediate import of presequence-containing proteins but also may directly be involved in the carrier import pathway. The fact that TimRhom I and TimRhom II, unlike in the case of the stuck LDH-DHFR, were not recovered with the arrested MCP12 variant indicates that they may be specifically required for the presequence pathway.

## Discussion

TbTim17 in *T. brucei* shows clear homology to the pore-forming core subunits of the TIM complexes in yeast. Its ablation confirms that the protein is essential for mitochondrial protein import. Thus, we used it as a bait to identify candidate subunits of the trypanosomal TIM machinery. Reciprocal IPs revealed 10 subunits that form the core of the complex. Besides TbTim17, the detected proteins included ACAD and TbTim62 that were previously reported to be associated with TbTim17 and the three novel components TbTim42, TimRhom I and TimRhom II. Furthermore, four tiny Tim-like proteins were also found in the complex. Other previously proposed subunits of the TIM complex in *T. brucei* such as TbTim50, TbTim47 and TbTim54 (refs 32,33) were neither found to interact with the three bait proteins, nor were they part of the complex containing the arrested LDH-DHFR.

According to the textbook, mitochondria have two distinct TIM translocases. One termed the TIM23 complex that imports presequence-containing proteins and the TIM22 complex specialized in inserting multi-spanning MCPs. The TIM23 and the TIM22 complexes do not share common subunits; however, the three central components of these complexes, namely Tim23, Tim17 and Tim22, are homologous and belong to the same protein family. Two of these proteins, Tim23 and Tim22, form the protein-conducting pores in their respective translocases[5,17]. The fact that in trypanosomatids TbTim17 is the only protein of the Tim17/22/23 family is surprising and suggests that in those cells it may mediate import of both presequence-containing proteins as well as MCPs.

All 10 identified TIM core subunits are components of the active presequence translocase as they were co-purified with a MTS-containing precursor protein that was arrested in the import channel. The isolated complex therefore functions analogous to the yeast TIM23 complex. However, a number of evidences suggest that many of its subunits may at the same time also mediate import of MCPs: (i) SILAC RNAi against TbTim17 resulted in a decrease of the steady-state levels of essentially all detected MCPs; (ii) tiny Tim-like chaperones, which in other organisms are exclusively associated with the translocation machineries for hydrophobic proteins, such as the TIM22 complex[19], are stable components of the *T. brucei* IM presequence translocase, (iii) IPs of TbTim42 and TbTim17 recovered three MCPs, which may represent substrates in transit to be inserted into the IM (Supplementary Fig. 1); and (iv) IP of a truncated tagged version of MCP12 that forms an intermediate that is stuck in the carrier import pathway recovers eight out of the 10 TIM subunits found in the presequence translocase.

In our study, we have discovered three novel integral membrane subunits of the trypanosomal presequence translocase, termed TbTim42, TimRhom I and TimRhom II. All are conserved in kinetoplastids but share no significant similarity with mitochondrial protein import components from other species. They constitute essential parts of the active presequence translocase as evidenced by their association with the LDH-DHFR translocation intermediate as well as the growth and protein import defects observed upon their ablation.

The identification of the rhomboid-like proteins TimRhom I and TimRhom II as essential functional components of the trypanosomal presequence translocase that act independently of TbTim17 is striking since some members of the rhomboid family were shown to be involved in protein translocation in other organelles. Rhomboid-like proteins of the derlin family are subunits, possibly even the translocation channel, of the endoplasmic reticulum-associated protein degradation (ERAD) system[50], which exports misfolded proteins from the ER lumen to the cytosol for degradation by the proteasome[51]. Moreover, in complex plastids of red algae, which are surrounded by four membranes, duplication or specialization of the ERAD system from the endosymbiont gave rise to a plastid-specific ERAD-like machinery, called SELMA, in their second outermost membrane[52]. This machinery was adapted such that protein translocation through the channel, which is likely formed by the two derlin orthologues, is no longer coupled to protein degradation[53].

Direct evidence that derlins function as protein-conducting pores is presently missing for any system. However, the fact that they lack proteolytic activity and that rhomboid-like proteins recognize and stabilize partially unfolded domains of their substrates at the interface between the membrane and the soluble phase would be consistent with a channel function[41,50,54].

It is presently unknown which proteins form the protein translocation pore(s) in the trypanosomal TIM complex(es). It has been shown that yeast Tim17, but not Tim22 or Tim23, can partially complement for the lack of TbTim17 in *T. brucei*[31]. This suggests that TbTim17 is a remote orthologue of Tim17 and therefore, as in yeast, does not form a protein-conducting channel on its own. It is tempting to speculate that TimRhom I and/or TimRhom II may be the functional analogue(s) of the yeast presequence translocation pore Tim23.

An alternative explanation for the presence of rhomboid-like proteins in the trypanosomal presequence translocase could be that they would clear the complex from stalled import substrates. However, the fact that TimRhom I and TimRhom II are predicted to lack protease activity and the fact that they are not only co-purified with the stalled LDH-DHFR precursor but also with the three tagged TIM subunits argues against this possibility.

Quite puzzling is the absence of the conserved PAM components Tim44 and mHsp70 (ref. 27) in the active presequence translocase even though orthologues for both proteins exist in *T. brucei*. It is possible that these more peripherally associated TIM complex components might have been lost during purification. However, taking into account the unusual composition of the trypanosomal presequence translocase and the presence of as yet uncharacterized mitochondrial proteins, including a putative AAA-ATPase, in the active translocase (Supplementary Data 5), it cannot be excluded that *T. brucei* utilizes a non-canonical motor complex for preprotein import.

Tim17, Tim22 and Tim23 show weak sequence similarity to the bacterial amino acid transport protein LivH, indicating that the ancestor of these proteins may have evolved from an amino acid permease that was present in the endosymbiont[55]. All other previously characterized TIM core subunits of other species do not show obvious similarity to bacterial proteins, suggesting they may have evolved in eukaryotes. The situation for the presequence translocase of *T. brucei* is quite different, since, besides TbTim17, its essential core includes the rhomboid-like proteins TimRhom I and TimRhom II. Rhomboid-like proteins are found in all three domains of life including α-proteobacteria[40]. Since TimRhom I and TimRhom II are not closer related to eukaryotic rhomboid-like proteins (for example, derlins, PARL) than to their bacterial counterparts, they may have

been commandeered from the original endosymbiont that gave rise to mitochondria.

Our results show that the essential core subunits of the *T. brucei* TIM complex(es) are highly diverged from the ones in other eukaryotes. Together with the recently discovered highly diverged trypanosomal analogue of the TOM translocase[28], these results reveal an unanticipated range of variation in the composition of mitochondrial protein import systems between different species. This is surprising since the general function of these systems to import more than 1,000 different proteins is conserved in all eukaryotes. Mitochondrial protein import was one of the first—if not the first—mitochondria-specific traits to evolve. Thus, mapping the compositional diversity of TIM and TOM complexes in different species may not only shed light into the early evolutionary history of eukaryotes, and thus help to reveal the early branches of the eukaryotic evolutionary tree but also disclose novel mechanisms of protein translocation across membranes.

## Methods

**Transgenic cell lines.** Transgenic *T. brucei* cell lines were engineered based on either procyclic strain 29-13 (ref. 56) or the F1γL262P variant of BSF strain 'New York single marker'[43]. They were cultivated at 27 °C in SDM-79 supplemented with 10% (v/v) fetal calf serum (FCS) or at 37 °C in HMI-9 containing 10% FCS, respectively.

C-terminal epitope tagging was done by fusing the full-length open reading frames (ORFs) of TbTim17, TimRhom I (Tb927.9.8260), TimRhom II (Tb927.8.4150), TbTim42 (Tb927.9.11220) and TbTim13 (Tb927.10.11520) (numbers appended to TbTim correspond to molecular weight) as well as the full-length and truncated (nt 274-912) ORF of MCP12 (Tb927.10.12840) to triple c-myc- or HA-tags. The LDH-DHFR fusion proteins used for the experiment in Fig. 5 and Supplementary Fig. 4 consist of the N-terminal 160 amino acids of the trypanosomal matrix protein dihydrolipoyl dehydrogenase (LDH) that were fused to mouse DHFR containing either a triple C-terminal HA- or c-myc-tag. The fragments encoding the tagged proteins were inserted into modified pLew100 vectors[56], in which the phleomycine resistance gene had been replaced by either the puromycine or the blasticidin resistance gene[57]. RNAi constructs were prepared using stem-loop inserts, the loop being a 460 bp spacer fragment. The resulting fragments were integrated into the same pLew100 vectors described above. The regions targeted by RNAi were: TimRhom I (ORF nt 364-669), TimRhom II (3′-untranslated region nt 125-364) and TbTim42 (ORF nt 110-642). The TbTim17 RNAi cell line has been described before[36]. A list of the primers used for the aforementioned constructs is provided in Supplementary Note 1.

**Antibodies.** Polyclonal rabbit antisera were commercially produced (Eurogentec, Belgium) using the indicated peptides as antigens: TbTim42, aa 133-148 (CADFVRFPPREQRFES) and aa 166-181 (CYKPSKVRSMRSPFEV); TimRhom I aa 91–105 (C + QRNEGDKGGDEEQKQ) and aa 68–83 (CSSAVLRDPKKPSGQL). For western blots (WB) the TbTim42 and the TimRhom I antisera were used at 1:20 and 1:500 dilutions, respectively. Their specificity was confirmed using WB of whole-cell extracts of uninduced and induced RNAi cell lines for both proteins (Supplementary Fig. 5). Commercially available antibodies were: mouse anti-c-myc (Invitrogen, Product No. 132500; dilution WB 1:2,000, IF microscopy (IF) 1:50); mouse anti-HA (Enzo Life Sciences AG, Product No. CO-MMS-101R-1000, dilution WB 1:5,000, IF 1:1,000); mouse anti-EF1a (Merck Millipore, Product No. 05-235, dilution WB 1:10,000). Antibodies previously produced in our laboratory are: polyclonal rabbit anti-voltage-dependent anion channel (dilution WB 1:1,000); polyclonal rabbit anti-ATOM40 (dilution WB 1:10,000, IF 1:1,000); polyclonal rabbit anti-CoxIV (dilution WB 1:1,000); polyclonal rabbit anti-Cyt C (dilution WB 1:100) and anti-Tim9 (dilution WB 1:20)[22,28]. Secondary antibodies used: goat anti-mouse IRDye 680LT conjugated (LI-COR Biosciences, P/N 926-68020, dilution WB 1:20,000, IF 1:100); goat anti-Rabbit IRDye 800CW conjugated (LI-COR Biosciences, P/N 926-32211, dilution WB 1:20,000); goat anti-rabbit FITC conjugated (Sigma, P/N F0382, dilution IF 1:100).

For IP of TimRhom I, 2 ml of rabbit anti-TimRhom I serum as well as the respective pre-immun serum were each allowed to bind for 2 h (RT) to 500 μl of GammaBind G Sepharose (GE Healthcare), which was pre-equilibrated in PBS. Unbound antibodies were removed by three washes in PBS. After equilibration in 100 mM sodium borate pH 9.0, the antibodies were crosslinked to the matrix by incubating two times for 30 min in fresh 20 mM dimethyl pimelimidate (in 100 mM sodium borate pH 9.0). Non-crosslinked antibodies were removed by two washes in 50 mM glycine pH 2.5 followed by five washes in PBS.

**Digitonin extraction.** Digitonin extraction was used to generate crude mitochondrial enriched fractions[58] for IP experiments, BN–PAGE analysis or to demonstrate mitochondrial localization of a protein of interest. For this, $1 \times 10^8$ cells were incubated for 10 min on ice in 20 mM Tris-HCl pH 7.5, 0.6 M sorbitol, 2 mM EDTA containing 0.015% (w/v) digitonin to selectively lyse the plasma membrane. After centrifugation (6,800 g, 4 °C), the resulting mitochondria-enriched pellet was separated from the supernatant, $2.5 \times 10^6$ cell equivalents of each fraction were subjected to SDS–PAGE and immunoblotting. Alternatively, the mitochondria-enriched pellet was used for further experiments as indicated below. For visualization of tagged proteins, the respective cell lines were induced for 24 h with tetracycline.

**Alkaline carbonate extraction.** A mitochondria-enriched pellet fraction obtained by digitonin extraction was resuspended in 100 mM $Na_2CO_3$ pH 11.5, incubated on ice for 10 min and centrifuged (100,000 g, 4 °C, 10 min) to separate the membrane fraction from soluble proteins. All samples were analysed by SDS–PAGE und immunoblotting. Equal cell equivalents were loaded for all fractions.

**SILAC proteomics and immunoprecipitation.** Cell lines allowing inducible RNAi or inducible expression of tagged proteins were washed with PBS and transferred into SDM-80 (ref. 59) containing 5.55 mM glucose and either light ($^{12}C_6/^{14}N_x$) or heavy ($^{13}C_6/^{15}N_x$) arginine (1.1 mM) and lysine (0.4 mM) (Cambridge Isotope Laboratories, USA). All cultures were grown in the presence of 10–15% dialysed FCS (BioConcept, Switzerland) for 6–10 doubling times to ascertain complete labelling of proteins with the heavy amino acids. For the TbTim17-RNAi cell line (Fig. 1), tetracycline induction was performed for 3 days. For SILAC-IP experiments, expression of the tagged proteins was induced for 1 day. Immediately before cell lysis, uninduced and induced cells were mixed in a 1:1 ratio. For TbTim17-SILAC RNAi experiments, digitonin-extracted crude mitochondrial fractions of the resulting mixed pellets were prepared and analysed by MS. All SILAC RNAi and IP experiments were performed in three biological replicates including a label-switch.

For the SILAC-IPs of tagged TbTim17, TbTim42 and TbTim13, digitonin-extracted crude mitochondrial fractions of $1–2 \times 10^8$ uninduced and induced cells each were solubilized for 15 min at 4 °C in 20 mM Tris-HCl, pH 7.4, 0.1 mM EDTA, 100 mM NaCl, 10% glycerol containing 1% (w/v) digitonin and 1 × Protease Inhibitor mix (EDTA-free, Roche). Following a clearing spin (20,000 g, 15 min, 4 °C), the lysate was transferred to affinity purification resin (30–60 μl EZview red anti-c-myc affinity gel from Sigma or 50–100 μl anti-HA affinity matrix from Roche) that had been equilibrated in wash buffer (20 mM Tris-HCl, pH 7.4, 0.1 mM EDTA, 100 mM NaCl, 10% glycerol containing 0.2% (w/v) digitonin). After 1 h of incubation at 4 °C, the supernatant was removed and the resin was washed three times with 500 μl wash buffer. To elute the bound proteins, the resin was boiled for 5 min in 60 mM Tris/HCl pH 6.8 containing 0.1% SDS. The resulting eluate was analysed by MS. SILAC-IP of the stalled carrier intermediate was essentially performed as described above. Instead of induced and uninduced cells, induced cells expressing either full-length MCP12 or the truncated version were differentially labelled and subjected to SILAC-IP. Elution was performed with 2% SDS in 60 mM Tris-HCl pH 6.8. SILAC-IP of the stalled import intermediate was performed as described above, except that 24 h before the IP, the SILAC-labelled cultures were supplemented with sulfanylamide (1 mM) and 500 μM aminopterine where indicated ( ± AMT). All cultures were induced for LDH-DHFR-HA expression 3 h before performing the IP. Elution was performed with 2% SDS in 60 mM Tris-HCl pH 6.8.

**Mass spectrometry and data processing.** Crude mitochondrial fractions prepared from TbTim17-SILAC RNAi cells were resuspended in urea buffer (30 mM Tris-HCl, 7 M urea, 2 M thiourea, pH 8.5) and proteins (4 μg protein per liquid chromatography/MS (LC/MS) analysis) were reduced, alkylated and tryptically digested in-solution as described before[60]. Sample preparation of proteins obtained in SILAC-IPs of tagged TbTim17, TbTim42 and TbTim13 including acetone precipitation, reduction, alkylation and tryptic in-solution digestion was performed as described[60]. For the MS analysis of proteins associated with the presequence import intermediate LDH-DHFR-HA or the carrier import intermediate MCP12-Δ1, immunoprecipitated proteins were separated by SDS–PAGE on a 4–12% NuPAGE BisTris gradient gel (Life Technologies). Proteins were stained using colloidal Coomassie Brilliant Blue and gel lanes cut into 12 or 10 slices, respectively. Reduction, alkylation, and tryptic in-gel digestion of proteins were performed as described before[61].

LC/MS analyses were performed using the UltiMate 3000 RSLCnano HPLC system (Thermo Scientific, Dreieich, Germany) directly connected to an LTQ-Orbitrap XL or Orbitrap Elite instrument (Thermo Scientific, Bremen, Germany). Peptides of the TbTim17-SILAC RNAi samples (analysed on the Orbitrap Elite; two technical replicates each) were separated applying a linear 265-min gradient ranging from 7.5% methanol (MeOH) and 4.5% acetonitrile (ACN) to 33.5% MeOH and 20% ACN in 0.1% formic acid (FA) at a flow rate of 250 nl min$^{-1}$. Simultaneous with the acquisition of MS survey scans (m/z 370–1,700) in the orbitrap at a resolution of 60,000 (at m/z 400), up to 25 of the most intense multiply charged precursor ions were further fragmented by collision-induced dissociation in the linear ion trap (TOP25 method). The dynamic exclusion time

preventing repeated fragmentation of precursor ions was set to 45 s. For the separation of peptide mixtures of the TbTim17, TbTim42 and TbTim13 IPs (analysed on the LTQ-Orbitrap XL), LC parameters were as follows: 2.4% MeOH/1.5% ACN—33.6% MeOH/21% ACN in 0.1% FA/4% DMSO in 130 min; flow rate, 250 nl min$^{-1}$. MS measurements were performed as described above applying a TOP5 method. For LC/MS analyses of the LDH-DHFR and MCP12 IPs (Orbitrap Elite), the following parameters were applied: 50-min gradient ranging from 0.5% MeOH/0.3% ACN to 31.2% MeOH/19.5% ACN in 0.1% FA/4% DMSO at a flow rate of 250 nl min$^{-1}$; acquisition of MS survey scans at resolution 120,000 ($m/z$ 400); TOP15 method; dynamic exclusion, 45 s.

MS raw data were processed for protein identification and SILAC-based relative quantification using MaxQuant/Andromeda[62,63] (version 1.3.0.5 for TbTim17 RNAi samples, v. 1.4.1.2 for TbTim42 and TbTim17 IPs, v. 1.5.1.0 for TbTim13 IPs, v. 1.5.3.12 for LDH-DHFR IPs, and v. 1.5.4.0 for MCP12 IPs). Tandem MS data were searched against all entries in the TriTryp database[64] (version 4.2 for TbTim17 RNAi data; version 7.0 for TbTim42 IP data; version 8.1 for the remaining data sets) using MaxQuant default parameters including Arg10 and Lys8 as heavy labels. Proteins were identified based on at least one unique peptide with a false discovery rate of 1% at the peptide and the protein level. Protein quantification was based on unique peptides and at least one SILAC peptide pair. The mean of log$_{10}$-transformed protein abundance ratios was calculated and a Student's $t$-test (two-sided for TbTim17 RNAi and MCP12 IP data; one-sided for all other data sets) was performed to determine $P$ values for all proteins quantified in at least two biological replicates. Proteins with a $P$ value < 0.05, a sequence coverage of at least 4% and a more than 1.5-fold downregulation (TbTim17 RNAi data) or a twofold enrichment (SILAC-IP data), respectively, were considered as candidate proteins in the respective experiments. For the LDH-DHFR-HA SILAC-IP results, shown in Fig. 5d, the data set was filtered for mitochondrial proteins, which were defined by all proteins that were detected by LC/MS in at least one of the two mitochondrial fractionation experiments as described[22].

**Immunoprecipitation for western blotting.** IPs destined for immunoblotting were essentially performed as described for the respective SILAC-IPs, simply omitting the differential labelling. IP of TimRhom I was performed in the same way using anti-TimRhom I antibodies and preimmune serum coupled to Sepharose (as described above) as precipitation matrix. Elution was in all cases performed using 2% SDS in 60 mM Tris-HCl pH 6.8 and samples were analysed by SDS–PAGE and western blotting.

**Immunofluorescence microscopy.** Expression of triple HA- or c-Myc-tagged TbTim42, TimRhom I and TimRhom II was induced for 24 h. Expression of LDH-DHFR-myc was induced overnight by addition of tetracycline in the presence of 1 mM sulfanylamide and in the presence or absence of 500 µM AMT as indicated. Cells were fixed with 4% paraformaldehyde in PBS and permeabilized with 0.2% Triton X-100 in PBS. Between the different incubation steps with primary and secondary antibodies, cells were washed with PBS. After postfixation in cold methanol, the slides were mounted with VectaShield containing 4′,6-diamidino-2-phenylindole (DAPI) (Vector Laboratories, P/N H-1200). Images were acquired with a DFC360 FX monochrome camera (Leica Microsystrems) mounted on a DMI6000B microscope (Leica Microsystems). Image analysis was done using LAS X software (Leica Microsystems).

For analysis of the mitochondrial membrane potential, uninduced and induced TbTim42, TimRhom I and TimRhom II RNAi cell lines were grown for 15 min in the presence of 500 nM MitoTracker Red CMXRos and then harvested and fixed as described before. As negative control served uninduced cells that were treated with 40 µM carbonyl cyanide m-chlorophenylhydrazone before incubation with Mitotracker.

**RNA extraction and northern blotting.** By acid guanidinium thiocyanate-phenol-chloroform extraction[65] total RNA was isolated from *T. brucei* and subsequently 7 µg was separated on a 1% agarose gel in 20 mM MOPS buffer, pH 7.0 containing 0.5% formaldehyde. Northern probes based on the RNAi inserts described above were prepared from gel-purified PCR products and radioactively labelled using the Prime-a-Gene labelling kit (Promega).

**Blue native–PAGE and 2D PAGE.** Mitochondria-enriched fractions were prepared by digitonin extraction as described above. After solubilization in 1% (w/v) digitonin (in 20 mM Tris-HCl pH 7.4, 50 mM NaCl, 10% glycerol, 0.1 mM EDTA) and a clearing spin, they were separated on 4–13% gradient gels for one-dimensional BN–PAGE and 6–16.5% BN–PAGE followed by 16% SDS–PAGE for two-dimensional PAGE. To facilitate transfer of protein complexes, two-dimensional PAGE gels were incubated in SDS–PAGE running buffer (25 mM Tris, 1 mM EDTA, 190 mM glycine, 0.05% (w/v) SDS) before western blotting.

**Data availability.** The mass spectrometry proteomics data have been deposited to the ProteomeXchange Consortium via the PRIDE [1] partner repository with the dataset identifier PXD005484. The remaining data are available within the article and its Supplementary Information files and from the corresponding author upon reasonable request. Full scans for northern and western blots are available in Supplementary Fig. 5. Molecular weight markers and the outlines of croppings shown in the main figures are indicated.

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

## Acknowledgements

A.H. gratefully acknowledges a fellowship from the Peter und Traudl Engelhorn foundation. We thank B. Knapp for technical assistance in LC-MS analyses. Research in the group of B.W. was funded by the Deutsche Forschungsgemeinschaft and the Excellence Initiative of the German Federal and State Governments (EXC 294 BIOSS Centre for Biological Signalling Studies). Research in the laboratory of A.S. was supported by grant 138355 and in part by the NCCR 'RNA & Disease' both funded by the Swiss National Science Foundation. Deposition of the data to the ProteomeXchange Consortium was supported by the PRIDE Team, EBI.

## Author contributions

A.H. and A.S. designed the experiments. A.H., C.W. and J.L.H. performed and analysed the experiments. Quantitative proteomic experiments were designed by S.O. and B.W. and SILAC-MS data were collected and analysed by S.O. A.S. and B.W. supervised the project. A.S. coordinated the entire project and obtained the main source of funding; A.H. and A.S. prepared and revised the manuscript with the input of S.O. and B.W.

## Additional information

**Competing financial interests:** The authors declare no competing financial interests.

