## [Peer Review File · Nature Communications]

Reviewer #1 (Remarks to the Author)

For their biogenesis, mitochondria import hundreds of precursor proteins. The import machinery of baker's yeast was studied over the last years in large detail. However, the composition and function of TOM and TIM complexes in other organisms is largely unknown. Even in mammalian mitochondria, detailed studies are missing and most conclusions are drawn by comparison of the similarity of gene sequences to sequences of yeast components.

In the present study the authors identified components of the TIM complex of Trypanosomes. Using the conserved and central Tim17 subunit as a bait, the authors identified a variety of potential TIM proteins. In particular, the authors showed that three of these proteins, TbTim42, TimRhom I and TimRhom II, are important for cell growth and mitochondrial protein import. Interestingly, the latter two are members of the rhomboid family of proteases. The authors speculate that these components were converted during evolution into translocation rather than degradation components.

This is an interesting study of high technical quality. The text reads very well and the data certainly support the conclusions. Even if Trypanosoma is an unconventional organism which might be only of direct relevance for a limited readership, the findings here are of rather general importance and certainly will inspire many to reevaluate the function of mitochondrial rhomboids in other organisms, including humans.

There are some minor points that should be considered:

1

The supplement contains a number of tables that will be of interest for many readers since they might want to search for homologs of TbTIM proteins in their specific model organisms. Unfortunately, the provided Protein IDs (TriTrypDB8.1) do not lead to any hits in the general protein databases such as that of NCBI. It would be good if the authors could make their protein hits available for the general readership of Nature Communications, also to increase the prospective citation rate of this study.

2

The observation that two rhomboids are part of TIM complex does not necessarily prove that these proteins directly contribute in protein import! It might be at least as likely that the rhomboids are recruited to the TIM complex in order to degrade stalled translocation intermediates. Depletion of such 'clearance factors' might lead to an accumulation of stalled proteins explaining the observed accumulation of the Cox4 precursor. The authors should discuss this possibility which still would be very interesting.

3

The two rhomboid proteins are very different to rhomboids of animal mitochondria or of bacteria, but also very different towards each other. Do the authors expect that both rhomboids are part of one complex or are there TIM complex which either contain TimRhomI or TimRhomII?

4

Along the same lines, does the simultaneous depletion of TimRhomI and TimRhomII lead to a much more severe Cox4 accumulation phenotype? This was expected if there were two parallel TIM channels containing partially redundant rhomboids.

5

The authors speculate about translocation mechanisms that are similar to that of the ERAD or SELMA systems. In both cases ubiquitination is essential to drive translocation. This is presumably not expected in mitochondria, or? Is there any indication for an ubiquitin-like modification in Trypanosome mitochondria? This also would explain why no ATP-dependent motor was identified.

6

The authors write: "The parasitic protozoon Trypanosoma brucei is only remotely related to yeast

and mammals. It is one of the earliest diverging eukaryotes that has a fully functional mitochondrion capable of oxidative phosphorylation." This sentence is misleading. Already the bacterial endosymbiont obviously had a functional respiratory chain. Many "primitive" eukaryotes have lost the ability to respire secondarily, and the group that lacks respiratory enzymes is therefore polyphyletic. The absence of a respiratory chain therefore must not be confused with primitiveness! It is rather a sign for being far derived from the origin!

Moreover, we know only very little about these "primitive" eukaryotes, so how can we know whether *Trypanosoma* is one of the earliest eukaryotes that has a functional respiration system? The authors should better remove this sentence or should be more cautious with the wording about evolutionary relationships.

Reviewer #2 (Remarks to the Author)

The manuscript by Harsman et al analyses the mitochondrial inner membrane protein translocase of *Trypanosoma brucei*. The mitochondrial protein import pathways in *T. brucei* appear to be considerably different from the pathways identified in yeast and in humans. The data available so far suggest that translocation across mitochondrial inner membrane of *T. brucei* is particularly peculiar. Yeast and humans have two separate inner membrane protein translocases to deal with mitochondrial proteins synthesized in the cytosol. The TIM23 complex deals with presequence-containing proteins and the TIM22 complex with the carrier proteins. Both complexes have members of the Tim17/22/23 protein family as their cores. In contrast, *T. brucei* appears to have only one member of this protein family, TbTim17, and the complex containing TbTim17 was proposed to import both classes of proteins. The molecular nature of TbTIM complex is still only poorly understood. In 2012, Singha et al used affinity purification via TAP-tagged TbTim17 coupled to MS analysis of CBB-stained proteins after SDS-PAGE, to identify TbTim47, TbTim54 and TbTim62 as the additional components of the TIM complex in *T. brucei* (Singha et al, JBC 2012). In the current manuscript, Harsman et al use a SILAC-based mass spectrometry approach to identify about 40 proteins enriched more than 2 fold in immunoprecipitates of myc-tagged TbTim17. The lists of identified interactors in two studies show some but not overly compelling overlap. Harsman et al then performed reciprocal SILAC-IP experiments using two interactors of TbTim17 they identified and named TbTimTim42 and TbTim13, as baits. Based on the three SILAC-IP experiments, the authors chose to further characterize 3 novel proteins, TbTim42, TimRhom I and TimRhom II, as putative novel components of the TIM complex. The authors convincingly show that all three proteins are mitochondrial membrane proteins whose depletion by RNAi arrests cell growth and leads to accumulation of the precursor form of CoxIV. Using a similar SILAC-IP experiment but now with a translocation intermediate stalled in the outer and inner membrane import channels as a bait, the authors show that the 3 identified proteins can also be enriched with the active translocase. TimRhom I and II are particularly appealing as novel components of the TIM complex as they appear to belong to a family of rhomboid-like protein that were previously proposed (but never directly shown) to form protein translocation pores in ERAD system as well as in complex plastids of red algae. Based on this notion, the authors propose that TimRhom I and II may contribute to pore formation in the *T. brucei* TIM complex. The manuscript is clearly written and addresses a very interesting topic that is not only important from the basic science point of view but may turn out to be very medically relevant in the future as well. Identification of rhomboid-like proteins as two novel subunits of the *T. brucei* TIM complex is certainly very appealing, however, in my opinion, not sufficiently supported by the data presented. Should the conclusions be better supported by the experimental data, a revised version of the manuscript may in principle be suited for a wide audience of Nature communications. My major concerns are detailed below.

1. Both rhomboid-like proteins were identified by SILAC-based MS in IPs of TbTim17, TbTim13 and TbTim42, one should say among many other candidates. I could not help but think that these were cherry picked from the list as they were not always among the most enriched proteins. In the discussion, the authors use the data from the very same IPs to actually conclude that some of the carrier proteins identified in the same IPs likely represent substrates in transit. Why is that argument not valid for rhomboid-like proteins as well? In addition, the authors state that the three

SILAC-IPs exhibited an overlap of 10 proteins including all three baits, however, I could not find TbTim17 on the list of proteins identified in IP using TbTim42 as a bait. It would be important to show CBB or silver stained gels of these various IPs and the control IPs so that the readers can evaluate by eye both the compositions of these various complexes and how stoichiometric the various interactions are. MS-based approaches are nowadays getting so sensitive that a likelihood of false positives is increasing with every new generation of mass spectrometers. Also, IPs using TimRhom I and II as baits should also be performed.

2. The authors use BNPAGE in Figure 3d to show that TbTim42 and a fraction of TimRhom I run in complexes of similar molecular mass as the TbTim17-containing complex. Unfortunately, the authors could not analyse TimRhom II in the same assay. In my opinion, this finding on its own is not a strong argument for either of the two newly identified proteins being constituents of the TIM complex - a finding that two proteins show a complex of the similar molecular mass is certainly not a proof that they are both part of the same complex as many unrelated protein complexes have similar molecular masses. It also appears that the majority of TimRhom I forms complexes of different molecular masses as compared to TbTim17 and TbTim42, though this could also be due to partial complex dissociation. It would be important to analyse whether depletions of TbTim42, TimRhom I and II lead to size reductions of TbTim17-containing complex and vice versa. The authors should also use antibody shift assays coupled to BNPAGE to show that the 700-1000kDa TbTim17-containing complex indeed contains all these newly identified proteins.

3. The findings that downregulation of TbTim42, TimRhom I and II arrest cell growth and lead to accumulation of CoxIV precursor, presented in Figure 4, also do not necessarily prove their direct involvement in the TIM complex. Protein import into mitochondria is a process sensitive to membrane potential and ATP levels and many proteins whose downregulation affects either of the two show a very similar phenotype. Mitochondrial protein import field has certainly witnessed such misinterpretations in the past.

4. In Figure 5, the authors use a chimeric MTS-containing precursor protein fused to DHFR to generate translocation intermediates and subsequently isolate the active translocase. This elegant assay has been very useful to analyze protein import into mitochondria and the authors show that, in the presence of aminopterin, all of the precursor shifts to a high molecular weight complex, as does the ATOM complex, translocase of the outer membrane. Yet, the SILAC-IP analysis of the active translocase isolated using the arrested precursor as a bait had to be "filtered for mitochondrial proteins because the enrichment of many non-mitochondrial proteins". Does this mean that most of the chimeric protein is not properly targeted to mitochondria? Or do all these other proteins associate with DHFR still exposed to the cytosol in the translocation intermediate? It would also be important to show that the TbTim42- and TimRhom I-containing complexes can similarly be shifted to higher molecular masses in the presence of aminopterin.

5. The authors could strengthen identification of novel interactors of TbTim17 as genuine components of the TIM complex by showing that they can be crosslinked to an arrested translocation intermediate.

6. I am wondering whether the experiment shown in Figure 1 can at all be used as an argument in to show that the single TIM complex is involved in translocation of both presequence-containing proteins and carrier proteins. The authors use a SILAC-MS-based quantitative proteomic approach to analyze steady-state levels of mitochondria-enriched fraction isolated from cells treated with TbTim17RNAi and control cells. Since they find that both presequence-containing and carrier proteins were reduced upon TbTim17 depletion, they conclude that this protein is involved in import of both types of proteins. Though I personally have little doubt that *T. brucei* has only one TIM complex, steady-state levels of mitochondrial proteins are not only affected by their synthesis and import but also by their degradation. Thus, strictly speaking, it is, in my opinion, impossible to differentiate between reduced import and increased degradation as the reason behind changed steady-state levels of proteins in the experiment shown in Figure 1, at least not in its current form. This is especially true since the experiment has been done at only one time point, 3 days, of TbTim17-RNAi treatment. How did the cell growth look like and how much was TbTim17 downregulated at this time point of treatment (at least I haven't been able to find TbTim17 quantification in the Sup. Table1)? The authors should address a possible contribution of increased degradation to changes in steady-state levels of various mitochondrial proteins - downregulation of

TbTim17 is, at a certain stage, likely going to have many pleiotropic effects on mitochondria and it is frequently observed that subunits of protein complexes are more easily degraded if their interaction partners are missing. The authors could analyse a possible contribution of increased degradation by quantifying for example subunits encoded in mtDNA as their levels should not be directly affected by absence of TbTim17. Has any of the subunits encoded in mtDNA been identified/quantified by SILAC-MS? Again, I haven't been able to find any of them in the Sup Table 1. Alternatively, a pulse-chase experiment coupled to SILAC-MS could be done.

7. I find the Supplementary Tables with MS quantification data very difficult to look at and interpret. The vast majority of proteins are listed with their protein IDs from the Trypanosoma database and described as "hypothetical protein, conserved". It would be helpful for a wider audience if human and/or yeast homologues were included in the Tables. I also wonder what "hypothetical protein, conserved" actually really means as for example all ATOM subunits are also listed with the same description. The authors themselves previously confirmed ATOM subunits as expressed proteins so I don't think that "hypothetical" is the right term to describe them. In contrast, the majority of them appear not to be conserved, at least not in yeast or humans, so I think it would be good if it would be mentioned somewhere to which organisms "conserved" refers to.

Reviewer #3 (Remarks to the Author)

The manuscript by Harsman et al. describes the characterization of the mitochondrial inner membrane translocase system in *T. brucei*. The authors results indicate that this translocase complex diverges from canonical systems, found in many other eukaryotic organisms, and contains two essential rhomboid-like proteins that haven't been associated with mitochondrial TIM complexes before. Moreover, they claim that *T. brucei* has only a single TIM complex, instead of two, that is able to translocate both mitochondrial presequence containing proteins and mitochondrial carrier proteins.

This work represents a significant contribution to the field of mitochondrial protein import in parasites, the earliest diverging eukaryotes, and improves our understanding on the evolution of such an important process to the biogenesis of present-day mitochondria.

The paper is well written and the results depicted are well presented and discussed. The conclusions are robust and well supported by the presented results. The authors give appropriate credit to previous work. I strongly support the publication of this paper in Nature Communications upon minor modifications.

1- The authors refer in "Materials and Methods" section that TbTim17 RNAi cell line has been described before (reference 36). However, a sentence should be added to the results section (next to line 99) describing the observed phenotypes of this strain. A western or a northern blot could also be included showing the decrease in expression of TbTim17 upon RNAi. This information will be important for the interpretation of the MS results.

For instance, is it possible that the 15% decrease observed in outer membrane proteins upon Tim17 ablation results from mitochondrial morphology alterations (secondary effect) and not from a decreased import of these proteins due to Tim17 depletion?

2- TbTim17 was knockdown and proteins from induced and uninduced parasites were identified. Why was not TbTim17 identified (not present in supplementary table 1)? Moreover, wouldn't you expect a higher downregulation of the other components of the TIM complex in TbTim17 RNAi induced strain? Does this observation suggests that TbTim17 is not needed for TIM assembly?

3-The authors stated that "TimRhom II could not be analyzed..." line 163 referring to Fig 3d. However, the immunoblot in this figure is from an SDS-containing gel and antibodies against tagged-TimRhom II were able to detect the protein in the SDS-PAGEs of Fig 3a and 3b. Did you try using myc-tagged TimRhom II expressing strains in first dimension BN-PAGE followed by

denaturing SDS-PAGE to detect TimRhom II? Moreover, upon TimRhom II RNAi (Fig 4b) expression of Tim17 is not affected. How do the authors reconcile these results?

4- In line 212 the authors conclude that "addition of AMT...". This sentence needs to be revised. The conclusion that the supercomplex is made of TOM and TIM complexes, although correct, is premature in here.

5- Parasites expressing LDH-DHFR-HA were used for IP experiments. However, parasites expressing LDH-DHFR-myc were used for IF. Why using different tagged- LDH-DHFR in different experiments?

6- To identify novel trypanosomal TIM complex subunits, the authors performed SILAC-based quantitative IP MS analysis. For this purpose induced and uninduced parasites were mixed and then samples for IP were prepared. My concern is on how the proteins from the uninduced parasites (lacking the tagged bait protein) will bind to the affinity purification resin in order to be purified and detected by MS.

About the statistical analysis, even though I am not a specialist, I think it was done correctly throughout the paper. The statistical tests are appropriated as well as the treatment of uncertainties.

Reviewer #1 (Remarks to the Author):

This is an interesting study of high technical quality. The text reads very well and the data certainly support the conclusions. Even if Trypanosoma is an unconventional organism which might be only of direct relevance for a limited readership, the findings here are of rather general importance and certainly will inspire many to reevaluate the function of mitochondrial rhomboids in other organisms, including humans.

There are some minor points that should be considered:

We thank the reviewer for his/her enthusiastic comments on our manuscript.

1. The supplement contains a number of tables that will be of interest for many readers since they might want to search for homologs of TbTIM proteins in their specific model organisms. Unfortunately, the provided Protein IDs (TriTrypDB8.1) do not lead to any hits in the general protein databases such as that of NCBI. It would be good if the authors could make their protein hits available for the general readership of Nature Communications, also to increase the prospective citation rate of this study.

We have now included the UniProt IDs of all detected proteins (as provided by the TriTrypDB, version 8.1) in all of our Supplementary tables. Moreover, the UniProt IDs in the tables are hyperlinked to the web pages of the corresponding proteins. Using the BLAST function on the UniProt web pages, it is now possible to find out very rapidly whether any of the listed trypanosomal proteins has orthologues in other species.

2. The observation that two rhomboids are part of TIM complex does not necessarily prove that these proteins directly contribute in protein import! It might be at least as likely that the rhomboids are recruited to the TIM complex in order to degrade stalled translocation intermediates. Depletion of such 'clearance factors' might lead to an accumulation of stalled proteins explaining the observed accumulation of the Cox4 precursor. The authors should discuss this possibility which still would be very interesting.

We agree that this possibility should be discussed in manuscript. However, we do not think the suggested scenario is very likely for the following reasons:

- i) The rhomboid-like proteins that associate with the trypanosomal TIM complex, as their counterparts in the ERAD and the SELMA systems, are predicted to be inactive and therefore could not degrade stalled proteins (compare Supplementary Fig. 2).
- ii) The rhomboid-like proteins are tightly associated with the TIM complex independent of whether the latter is isolated by pull downs of its subunits (Fig. 2) or with the help of a stalled presequence-containing precursor protein (LDH-DHFR, Fig. 5).

We have added the following paragraph in the discussion section (lines 327-331) of the revised manuscript.

"An alternative explanation for the presence of rhomboid-like proteins in the trypanosomal TIM complex could be that they would clear the translocase from stalled import substrates. However, the fact that TimRhom I and TimRhom II are predicted to lack protease activity and the fact that they are not only co-purified with the stalled LDH-DHFR precursor but also with the three tagged TIM subunits argues against this possibility."

3. The two rhomboid proteins are very different to rhomboids of animal mitochondria or of bacteria, but also very different towards each other. Do the authors expect that both rhomboids are part of one complex or are there TIM complex which either contain TimRhomI or TimRhomII?

The ERAD and the SELMA system each contain two distinct rhomboid-like proteins. Thus, we find the idea that the two rhomboid-like proteins detected by us in four independent immunoprecipitations are present in a single trypanosomal TIM complex attractive. However, presently we do not have any experimental data which could exclude the existence of two TIM complex variants that each would be associated with only one of the two rhomboid-like proteins.

4. Along the same lines, does the simultaneous depletion of TimRhomI and TimRhomII lead to a much

more severe Cox4 accumulation phenotype? This was expected if there were two parallel TIM channels containing partially redundant rhomboids.

The suggested experiment would in principle be feasible. However, based on our experience with other proteins we do not think it would be very informative. The RNAi cell lines against the two rhomboid-like protein already show a very strong growth phenotype that starts very early after induction of RNAi (especially for TimRhom I). Thus, in a combined RNAi cell line the cells will likely die even more rapidly. Thus, it would be very difficult to distinguish specific biochemical phenotypes from pleiotropic effects caused by the expected rapid cell death.

5. The authors speculate about translocation mechanisms that are similar to that of the ERAD or SELMA systems. In both cases ubiquitination is essential to drive translocation. This is presumably not expected in mitochondria, or? Is there any indication for an ubiquitin-like modification in Trypanosome mitochondria? This also would explain why no ATP-dependent motor was identified.

This is a very interesting question. Should import of proteins across the inner membrane in trypanosomes indeed be linked to ubiquitination, the modification would need to be transient as there is no evidence that trypanosomal mitochondrial proteins are constitutively ubiquitinated. If transient ubiquitination would occur we would expect LDH-DHFR, the stalled substrate protein in the experiment shown in Fig. 5, to accumulate in its ubiquitinated form. However, in our MS analysis we detected the LDH-DHFR substrate but we did not find a significant aminopterin-dependent enrichment of ubiquitin-like proteins. Based on these results we favor the idea that ubiquitination is not involved in the import process.

6. The authors write: "The parasitic protozoon Trypanosoma brucei is only remotely related to yeast and mammals. It is one of the earliest diverging eukaryotes that has a fully functional mitochondrion capable of oxidative phosphorylation." This sentence is misleading. Already the bacterial endosymbiont obviously had a functional respiratory chain. Many "primitive" eukaryotes have lost the ability to respire secondarily, and the group that lacks respiratory enzymes is therefore polyphyletic. The absence of a respiratory chain therefore must not be confused with primitiveness! It is rather a sign for being far derived from the origin!

Moreover, we know only very little about these "primitive" eukaryotes, so how can we know whether Trypanosoma is one of the earliest eukaryotes that has a functional respiration system? The authors should better remove this sentence or should be more cautious with the wording about evolutionary relationships.

We agree that the phrase might be misleading. We do not want to imply that non-respiring mitochondria are more ancestral than respiring ones. What we mean is, that *T. brucei* is considered by many to be a representative of one of the earliest - if not the earliest - diverging branch of the eukaryotic evolutionary tree. Independent of that *T. brucei* has mitochondria that in functional terms are comparable to the ones of yeast and mammals. We changed the phrase slightly, it now reads: "It is one of the earliest diverging eukaryotes and has a fully functional mitochondrion capable of oxidative phosphorylation." (line 66)

Reviewer #2 (Remarks to the Author):

The manuscript is clearly written and addresses a very interesting topic that is not only important from the basic science point of view but may turn out to be very medically relevant in the future as well. Identification of rhomboid-like proteins as two novel subunits of the T. brucei TIM complex is certainly very appealing, however, in my opinion, not sufficiently supported by the data presented. Should the conclusions be better supported by the experimental data, a revised version of the manuscript may in principle be suited for a wide audience of Nature communications. My major concerns are detailed below.

We appreciate that the reviewer thinks that our manuscript deals with a very interesting topic. We are convinced the additional experimental analyses presented in the revised manuscript strengthen our conclusions.

1. Both rhomboid-like proteins were identified by SILAC-based MS in IPs of TbTim17, TbTim13 and

TbTim42, one should say among many other candidates. I could not help but think that these were cherry picked from the list as they were not always among the most enriched proteins. In the discussion, the authors use the data from the very same IPs to actually conclude that some of the carrier proteins identified in the same IPs likely represent substrates in transit. Why is that argument not valid for rhomboid-like proteins as well?

All TIM subunits, including the two rhomboid-like proteins, are greatly enriched in all three SILAC-IPs (using tagged *TbTim17*, *TbTim13* and *TbTim42*), although *TbTim17* in the *TbTim42* IP is only detected on immunoblots (discussed below). The three carrier proteins, thought to be in transit in the TIM complex, however, are only poorly enriched in two out of three IPs (This indicated with green and blue colors in Supplementary Fig. 1). - Finally, in contrast to all TIM subunits, none of the carrier proteins is significantly enriched in the IPs of tagged LDH-DHFR that is stalled in the import pore (Fig. 5).

In addition, the authors state that the three SILAC-IPs exhibited an overlap of 10 proteins including all three baits, however, I could not find TbTim17 on the list of proteins identified in IP using TbTim42 as a bait.

The reviewer is correct, unlike claimed in the original manuscript *TbTim17* is not detected by MS in the SILAC pull down analysis of tagged *TbTim42*. We apologize for this mistake. We know that *TbTim17* is difficult to detect by MS, which also explains why it was not detected in the global RNAi analysis shown in Fig. 1a. However, Figure 1 below (for review only) illustrates that *TbTim17* can be detected on immunoblots from SDS gels showing that it is efficiently recovered in the pull down experiments with the tagged *TbTim42*. This indicates that its absence in the SILAC-RNAi or SILAC-CoIP MS analyses is due to technical reasons.

Figure 1 (for review only)

Pull down of HA-tagged *TbTim42* efficiently recovers *TbTim17*

We have modified the legend of Fig. 2 (lines 732 - 734) to "Depicted are the three bait proteins, which reciprocally interact as shown by SILAC-IPs or by immunoblot, as well as all proteins specifically enriched in all three CoIPs"

It would be important to show CBB or silver stained gels of these various IPs and the control IPs so that the readers can evaluate by eye both the compositions of these various complexes and how stoichiometric the various interactions are. MS-based approaches are nowadays getting so sensitive that a likelihood of false positives is increasing with every new generation of mass spectrometers.

It is true that MS is getting more and more sensitive. However, this sensitivity is essential for most of our experiments. While all methods to quantify the enriched proteins in an IP have their disadvantages, we believe that the combination of SILAC with MS provides the best solution to quantify the specific enrichment of proteins in the eluate fractions and to distinguish specific binding from background binding. The SILAC methodology allows mixing of the control and the experimental samples prior to the IP reaction. This means that - by definition - both samples are treated absolutely identical. To provide a stained gel lane of the "mixed" eluates from SILAC-IP experiments does therefore not make sense.

To do the suggested experiments we would need to repeat all IP reactions using the standard protocol, in which the control and the experimental samples are processed separately. However, the experimental variation of such experiments would be much bigger. Furthermore, to see individual bands on Coomassie-stained gels we would need to upscale the IPs quite dramatically which would not be practical. Using silver-stained gels might be more realistic. However, we do not think this

would allow to judge the stoichiometry of the immunoprecipitated components, as it is known that the dynamic range of silver-stained gels is very low. Furthermore, the staining intensity of many silver-stained proteins is an intrinsic function of specific proteins and some proteins do not stain at all (Steinberg, *Methods Enzymol.* 2009, 464: 541-563).

Finally, we would like to emphasize that all our SILAC-IP analyses were performed in triplicates. Thus, for each of the detected proteins the mean of the enrichment factor and the corresponding p-value is provided.

Also, IPs using TimRhom I and II as baits should also be performed.

We wanted to do these experiments. The problem however is that only a relatively small fraction of the tagged TimRhom I and TimRhom II were present in the TIM complex as evidenced by BN-PAGE. This suggests that the tag on TimRhom I and TimRhom II may interfere with the assembly of the protein into the TIM complex or its stability which in turn would make the interpretation of the suggested IPs very difficult.

Instead we performed an IP using our antiserum that recognizes native TimRhom I. The antiserum was incubated with protein G coupled Sepharose and the bound anti-TimRhom I IgGs were crosslinked to protein G. Figure 2 (for review only) shows that TbTim17 get depleted from the unbound fraction and is efficiently co-precipitated with TimRhom I, whereas the control protein cytochrome c1, is not recovered in the eluate. This experiment reinforces the notion that TimRhom I is a subunit of the trypanosomal presequence translocase. However, due to the distortions seen in the gel, which are due to the fact that despite the cross-linking and a preelution step much of the bound antibodies were co-eluted, we were not able to analyze the coprecipitation of the other complex subunit and thus would not like to include this result in the manuscript.

Figure 2 (for review only)

Pulldown of native TimRhom I efficiently recovers TbTim17

2. The authors use BN-PAGE in Figure 3d to show that TbTim42 and a fraction of TimRhom I run in complexes of similar molecular mass as the TbTim17-containing complex. Unfortunately, the authors could not analyse TimRhom II in the same assay. In my opinion, this finding on its own is not a strong argument for either of the two newly identified proteins being constituents of the TIM complex - a finding that two proteins show a complex of the similar molecular mass is certainly not a proof that they are both part of the same complex as many unrelated protein complexes have similar molecular masses.

We agree and we toned down the paragraph in question (lines 161-169). - However, we would like to stress that our claim that TbTim42, TimRhom I and TbRhom II are part of the same complex is supported by but not exclusively based on the BN-PAGE analysis. Much stronger support comes from the observed overlaps between: i) IPs using three different putative TIM subunits and ii) an IP of the arrested LDH-DHFR import intermediate. Moreover, our claim is corroborated by the fact that ablation of the putative TIM complex subunits leads to a protein import phenotype.

It also appears that the majority of TimRhom I forms complexes of different molecular masses as

compared to *TbTim17* and *TbTim42*, though this could also be due to partial complex dissociation.

Indeed we favor the idea that this might be due to partial complex dissociation.

*It would be important to analyse whether depletions of *TbTim42*, *TimRhom I* and *TimRhom II* lead to size reductions of *TbTim17*-containing complex and vice versa.*

We did these experiments. However, ablation of *TbTim42* did not lead to the accumulation of a low molecular weight complex but rather caused a rapid reduction of the whole *TbTim17*-containing complex (see Fig. 3, for review only, left panel), suggesting the *TbTim42* is required for its assembly. The same was seen after ablation of *TimRhom I* and *TimRhom II*, respectively, although the kinetics of reduction of the *TbTim17*-containing complexes was slower (see Fig. 3, for review only, middle and right panel). - The reciprocal experiments did not yield meaningful results for the following reasons.

Our antisera that recognize the native *TbTim42*, *TimRhom I* and *TimRhom II* proteins do not work on immunoblots from BN-PAGE. Tagging the proteins didn't help much either, since in the case of *TimRhom I* and *TimRhom II*, as explained above, the tag likely interferes with the assembly of the proteins into the TIM complex. Tagged *TbTim42* is efficiently assembled into the TIM complex and ablation of *TbTim17* indeed caused a reduction of the *TbTim42*-containing complex albeit with slower kinetics than in the reverse experiment (not shown).

Figure 3 (for review only)

BN-gels of inducible RNAi cell lines for *TbTim42*, *TimRhom I* and *TimRhom II* expressing HA-tagged *TbTim17*-HA (time of induction is indicated) were analyzed on immunoblots using anti-HA antibodies. A section of the Coomassie-stained gel is shown as a loading control.

*The authors should also use antibody shift assays coupled to BN-PAGE to show that the 700-1000kDa *TbTim17*-containing complex indeed contains all these newly identified proteins.*

This would be a direct way to show which proteins are in which complex. However, most of the antibodies we used only work if they are affinity purified on the peptides that were used to raise them. As a result the signals on the immunoblots are generally quite weak. The affinity-purified antisera are therefore not sufficiently concentrated to induce a quantitative shift of complexes on blue native gels. Moreover, the TIM complex is very big, even on 4-13% gel it doesn't migrate very far into the gel. It would therefore be difficult to see the expected antibody-induced size difference of the TIM complex.

We tried to do another experiment that in part addresses the same question (see comments to point 4 below)

*3. The findings that downregulation of *TbTim42*, *TimRhom I* and *TimRhom II* arrest cell growth and lead to accumulation of *CoxIV* precursor, presented in Figure 4, also do not necessarily prove their direct involvement in the TIM complex. Protein import into mitochondria is a process sensitive to membrane potential and ATP levels and many proteins whose downregulation affects either of the two show a very similar phenotype. Mitochondrial protein import field has certainly witnessed such misinterpretations in the past.*

We agree. Using the potential-sensitive dye Mitotracker we have now tested whether the membrane potential is still intact in the three induced RNAi cell lines (*TbTim42*, *TimRhom I* and *TimRhom II*) (Supplementary Fig. 3 in the revised manuscript). The results show that in all cases the accumulation of *CoxIV* precursor was observed at a time when the membrane potential was still intact. As a

negative control the cells were treated with CCCP, which - as expected - disrupted the membrane potential in all cases.

The following paragraph was inserted into the revised manuscript

Results (lines 190-191)

"The Mitotracker staining in Supplementary Fig. 3 shows that in all three cell lines the membrane potential was still intact at this time point."

4. In Figure 5, the authors use a chimeric MTS-containing precursor protein fused to DHFR to generate translocation intermediates and subsequently isolate the active translocase. This elegant assay has been very useful to analyze protein import into mitochondria and the authors show that, in the presence of aminopterin, all of the precursor shifts to a high molecular weight complex, as does the ATOM complex, translocase of the outer membrane. Yet, the SILAC-IP analysis of the active translocase isolated using the arrested precursor as a bait had to be "filtered for mitochondrial proteins because the enrichment of many non-mitochondrial proteins". Does this mean that most of the chimeric protein is not properly targeted to mitochondria? Or do all these other proteins associate with DHFR still exposed to the cytosol in the translocation intermediate?

It is true that for the graph shown in Fig. 5D we filtered the results for mitochondrial proteins. However, Supplementary Table 5 depicts all detected proteins, irrespectively of whether they were present in the mitochondrial proteome or not. Among the 41 proteins that are 2-fold or more enriched only 15 are non-mitochondrial and thus removed by our filter. The term "many non-mitochondrial proteins" in the original manuscript is therefore misleading, we replaced it by "a few non-mitochondrial proteins" (line 235-236).

In the presence of aminopterin a fraction of the precursor LDH-DHFR accumulates indeed in the cytosol (Supplementary Fig. 4a). However, we find it unlikely that the detected non-mitochondrial proteins in the aminopterin-treated sample are binding to this fraction or to the cytosol-exposed DHFR moiety of the stalled fusionprotein. We rather think that these proteins are detected as a consequence of the aminopterin treatment itself. Aminopterin is expected to inhibit the endogenous trypanosomal DHFR and thus will affect the physiology of the cell which may include the expression of novel proteins. Interestingly, 4 out of the 15 upregulated non-mitochondrial proteins are kinetochore components, suggesting that aminopterin disturbs the trypanosomal cell cycle.

It would also be important to show that the TbTim42- and TimRhom I-containing complexes can similarly be shifted to higher molecular masses in the presence of aminopterin.

We tried this experiment without a conclusive result. The problem was likely that, as mentioned above, the TIM complex is already very big (bigger than the ATOM complex) and as a consequence it doesn't migrate very far into the gel. Thus, the expected migration of the even bigger LDH-DHFR-HA ATOM-TIM supercomplex would be difficult to distinguish from the one of the TIM complex alone.

Instead we did a different experiment which we believe addresses at least in part the same question. We showed that the formation of the aminopterin-dependent ATOM-TIM supercomplex containing the stuck LDH-DHFR-HA depends on the presence of TimRhom I and TimRhom II. Figure 4 (for review only) shows that while the aminopterin-dependent LDH-DHFR-HA-containing complex is still formed after one day of tetracycline induction its formation is completely abolished after 3 days of induction.

Figure 4 (for review only)

Formation of the LDH-DHFR-HA TIM-TOM supercomplex depends on TimRhom I and TimRhom II. BN-gel of the TimRhom I and TimRhom II RNAi cell lines which constitutively express LDH-DHFR-HA were induced in the presence of aminopterin for 1 and 3 days, respectively, and subsequently analyzed by immunoblots using anti-HA antibodies.

These results are consistent with the notion that TimRhom I and TimRhom II are essential components of the presequence translocase. However, as indirect effects cannot entirely be excluded in this experiment we prefer not to add it to the revised manuscript.

5. The authors could strengthen identification of novel interactors of TbTim17 as genuine components of the TIM complex by showing that they can be crosslinked to an arrested translocation intermediate.

It is not entirely clear to us what the reviewer means. We are able to recover the TIM and ATOM complex subunits by precipitation of the stalled substrate without the need for cross-linking. This is in our view more convincing than if cross-linking would have been required. - We agree that cross-linking of the isolated TIM complex with the stalled intermediate has great potential to identify the pore subunits that directly interact with the stalled import substrate. However, as we do not claim to know the identity of the import pore, we think this would be beyond the scope of the present study.

6. I am wondering whether the experiment shown in Figure 1 can at all be used as an argument in to show that the single TIM complex is involved in translocation of both presequence-containing proteins and carrier proteins.

The reviewer questions whether the observed reduction of the steady state levels of presequence-containing and carrier proteins upon Tim17 RNAi is a convincing argument that a single TIM complex mediates import of both type of proteins.

We agree and have addressed this problem experimentally by producing an import intermediate for a carrier protein. Subsequent IPs were used to analyze whether the stuck carrier protein is associated with subunits of the TIM complex. The experiment worked very nicely, in order to include it in the manuscript we added a new Figure with the corresponding legend (Figure 6) and a new paragraph to the result section (see below). Moreover the discussion section was also adapted.

Results (lines 249-266)

"TIM subunits present in the carrier translocase

The proteomic analysis of the RNAi cell line in Fig. 1 suggests that TbTim17 might not only be involved in import of presequence-containing proteins but also in the biogenesis of MCPs. In order to investigate whether the trypanosomal TIM complex plays a direct role in import of MCPs we wanted to produce an import intermediate that is stuck in the carrier import pathway. MCPs consist of three tandemly repeated structurally similar modules each of which contains two transmembrane domains. Previous work in yeast has shown that a variant of the carboxylate carrier lacking the first module can still be imported across the outer membrane but becomes stuck at the TIM22 complex⁴⁹. Fig. 6 shows that the same strategy to produce an import intermediate in the carrier import pathway also works in trypanosomes. C-terminally myc-tagged full length MCP12 or a variant thereof lacking the first module were expressed in *T. brucei* (Fig. 6a). Both proteins localize to mitochondria (Fig. 6b) and according to carbonate extraction are integral membrane proteins (Fig. 6c). IPs using anti-myc antibodies shows that both tagged MCP12 variants are efficiently recovered in the eluate fractions (Fig. 6d). However, the TIM complex subunits TbTim17, TbTim42 and Tim9 specifically co-precipitated with the truncated variant of MCP but not with the full length protein. These results suggest that these TIM subunits do not only mediate import of presequence-containing proteins but may also directly be involved in the carrier import pathway. The fact that TimRhom I, unlike in the case of the stuck LDH-DHFR, was not recovered with the stuck MCP12 variant indicates that it may be specifically required for the presequence pathway."

The authors use a SILAC-MS-based quantitative proteomic approach to analyze steady-state levels of mitochondria-enriched fraction isolated from cells treated with TbTim17RNAi and control cells. Since they find that both presequence-containing and carrier proteins were reduced upon TbTim17 depletion, they conclude that this protein is involved in import of both types of proteins. Though I personally have little doubt that T. brucei has only one TIM complex, steady-state levels of

mitochondrial proteins are not only affected by their synthesis and import but also by their degradation.

We have used the reduction of the steady state levels of mitochondrial proteins as evidence for the inhibition of mitochondrial protein import. This is based on the observation that for a subset of imported proteins we not only see reduction of the mitochondrial steady state levels but simultaneous accumulation of the unprocessed precursor in the cytosol. We therefore assume that, at least for presequence-containing proteins, the reduction in the mitochondrial steady-state levels, which is seen upon ablation of many different import factors, including TbTim17, is indeed due to diminished import.

However, it is possible that for some proteins the reduction in the mitochondrial abundance could be a secondary effect. Thus, the reviewer is correct that we cannot know whether the reduction of the mitochondrial steady state levels of mitochondrial carrier proteins is because they directly need TbTim17 to be imported or because components of a hypothetical import machinery for carrier proteins are not correctly inserted into the inner membrane. Our novel experiments based on the MCP12 variants however confirm an interaction of TbTim17 with a mitochondrial carrier import intermediate and thus agree with a direct function of TbTim17 in carrier import.

Thus, strictly speaking, it is, in my opinion, impossible to differentiate between reduced import and increased degradation as the reason behind changed steady-state levels of proteins in the experiment shown in Figure 1, at least not in its current form. This is especially true since the experiment has been done at only one time point, 3 days, of TbTim17-RNAi treatment. How did the cell growth look like and how much was TbTim17 downregulated at this time point of treatment (at least I haven't been able to find TbTim17 quantification in the Sup. Table1)?

As mentioned above TbTim17 is difficult to detect by MS and was not detected in the experiment shown in Fig. 1a. We have now added a growth curve of uninduced and induced TbTim17-RNAi cell line grown in the SILAC medium and an immunoblot which illustrates the extent of the RNAi-induced downregulation of TbTim17 (see Fig. 1a of the revised manuscript). - The legend to Fig. 1 was changed accordingly.

The authors should address a possible contribution of increased degradation to changes in steady-state levels of various mitochondrial proteins - downregulation of TbTim17 is, at a certain stage, likely going to have many pleiotropic effects on mitochondria and it is frequently observed that subunits of protein complexes are more easily degraded if their interaction partners are missing. The authors could analyse a possible contribution of increased degradation by quantifying for example subunits encoded in mtDNA as their levels should not be directly affected by absence of TbTim17. Has any of the subunits encoded in mtDNA been identified/quantified by SILAC-MS? Again, I haven't been able to find any of them in the Sup Table 1. Alternatively, a pulse-chase experiment coupled to SILAC-MS could be done.

Most primary transcripts of mitochondrially encoded proteins in *T. brucei* are subject to extensive RNA editing consisting mainly of insertion of a variable number of uridine residues. This leads to a high frequency of UUU codons that specify phenylalanine. Thus, mitochondrially encoded proteins in trypanosomes are very hydrophobic, prone to aggregation and very difficult to detect by either gel electrophoresis or by MS (Horvath et al. Science. 2000, 287: 1639–1640; Škodová-Sveráková et al. 2015, 201:135-138).

However, we agree that the reduction of the steady state levels of mitochondrial proteins seen in induced Tim17-RNAi cells could in some cases also be due to increased protein degradation. We have therefore added the following phrase to the result section of the revised manuscript.

Results (lines 106-108)

"The observed reduction in the abundance of mitochondrial protein is likely mainly due to inhibition of mitochondrial protein import, although increased degradation of organellar proteins that lack stoichiometric amounts of cognate binding partners may contribute to it."

7. I find the Supplementary Tables with MS quantification data very difficult to look at and interpret. The vast majority of proteins are listed with their protein IDs from the Trypanosoma database and described as "hypothetical protein, conserved". It would be helpful for a wider audience if human

and/or yeast homologues were included in the Tables. I also wonder what "hypothetical protein, conserved" actually really means as for example all ATOM subunits are also listed with the same description. The authors themselves previously confirmed ATOM subunits as expressed proteins so I don't think that "hypothetical" is the right term to describe them. In contrast, the majority of them appear not to be conserved, at least not in yeast or humans, so I think it would be good if it would be mentioned somewhere to which organisms "conserved" refers to.

To make our data provided in the supplemental tables more accessible to a broader readership, we have now included the UniProt IDs for all proteins listed and added a hyperlink to the website of a given protein (see reply to Reviewer 1). It is not practical to provide information about human and/or yeast homologs for all trypanosomal proteins identified, but using the BLAST function on a protein's UniProt webpage makes it possible with two clicks only to find homologous proteins in other species.

The protein descriptions given in the tables have been extracted from the *Trypanosoma brucei* database (TriTrypDB, version 8.1). In the modified versions of the Supplementary tables, we updated the information given for protein description according to the latest entries in the TriTrypDB. In addition, we added information about the Gene Names (if available). The term "hypothetical protein, conserved" is widely used in databases, it is used for an ORF that encodes for a protein of unknown function that has homologs in other phylogenetic groups. In the case of trypanosomes the "hypothetical, conserved ORFs" are in essentially all cases restricted to the Kinetoplastids.

Reviewer #3 (Remarks to the Author):

This work represents a significant contribution to the field of mitochondrial protein import in parasites, the earliest diverging eukaryotes, and improves our understanding on the evolution of such an important process to the biogenesis of present-day mitochondria. The paper is well written and the results depicted are well presented and discussed. The conclusions are robust and well supported by the presented results. The authors give appropriate credit to previous work. I strongly support the publication of this paper in Nature Communications upon minor modifications.

We would like to thank the reviewer for her/his very positive evaluation of our study.

1- The authors refer in "Materials and Methods" section that TbTim17 RNAi cell line has been described before (reference 36). However, a sentence should be added to the results section (next to line 99) describing the observed phenotypes of this strain. A western or a northern blot could also be included showing the decrease in expression of TbTim17 upon RNAi. This information will be important for the interpretation of the MS results.

As explained in the comments to reviewer 2 we have now added a growth curve of uninduced and induced TbTim17-RNAi cell line grown in SILAC medium and an immunoblot which illustrates the extent of the RNAi-induced downregulation of TbTim17 (see Fig. 1a of the revised manuscript). Moreover the manuscript was modified as follows:

Result (lines 101-103)

"Three days after induction of RNAi equal cell numbers of uninduced and induced cultures were mixed and mitochondria-enriched fractions were prepared for further analysis by quantitative MS. At this time point the induced cells did not show a growth phenotype yet (Fig. 1a)"

For instance, is it possible that the 15% decrease observed in outer membrane proteins upon Tim17 ablation results from mitochondrial morphology alterations (secondary effect) and not from a decreased import of these proteins due to Tim17 depletion?

The morphology is not really changed after 3 days of induction of TbTim17 RNAi (in SILAC medium). - We do not know why the levels of a few OM proteins are decreased.

2- TbTim17 was knockdown and proteins from induced and uninduced parasites were identified. Why was not TbTim17 identified (not present in supplementary table 1)?

Based on previous analyses we know that TbTim17 is difficult to detect by MS.

Moreover, wouldn't you expect a higher downregulation of the other components of the TIM complex in TbTim17 RNAi induced strain? Does this observation suggest that TbTim17 is not needed for TIM assembly?

The SILAC-Tim17 RNAi MS analysis shown in Fig. 1a was done at early times of induction (3 days) which is before the growth phenotype appears. This may explain why an extensive ablation of the other TIM complex subunits is not apparent yet.

3- The authors stated that "TimRhom II could not be analyzed..." line 163 referring to Fig 3d. However, the immunoblot in this figure is from an SDS-containing gel and antibodies against tagged-TimRhom II were able to detect the protein in the SDS-PAGEs of Fig 3a and 3b. Did you try using myc-tagged TimRhom II expressing strains in first dimension BN-PAGE followed by denaturing SDS-PAGE to detect TimRhom II? Moreover, upon TimRhom II RNAi (Fig 4b) expression of Tim17 is not affected. How do the authors reconcile these results?

We tried these experiments. However, the problem was that only a small fraction of the tagged version of TimRhom II (as well as tagged TimRhom I) was detected in the TIM complex when analyzed by BN- PAGE. This suggests that tagged TimRhom II does not very efficiently assemble into the TIM complex or that the TIM complex containing tagged TimRhom II is not stable during BN PAGE. This interpretation is supported by the fact that in cell lines, which simultaneously express untagged and tagged TIM complex subunits, the aminopterin-induced LDH-DHFR ATOM-TIM complex becomes depleted of the tagged rhomboid subunits (not shown).

Moreover, upon TimRhom II RNAi (Fig 4b) expression of Tim17 is not affected. How do the authors reconcile these results?

It is correct that the steady state levels of Tim17 as measured by SDS PAGE are not affected during TimRhom II RNAi. However, when analyzed by BN-PAGE downregulation of TimRhom II causes a decline of the TbTim17 containing complexes suggesting that TimRhom II is required for TIM complex assembly. See comments reviewer 2, point 2 (Figure 3, for review only)

4-In line 212 the authors conclude that "addition of AMT...". This sentence need to be revised. The conclusion that the supercomplex is made of TOM and TIM complexes, although correct, is premature in here.

We modified the phrase: "Thus, addition of AMT induces the formation of a supercomplex consisting of the import substrate, the ATOM and the TIM complexes." as follows " Thus, addition of AMT induces the formation of a supercomplex, which as shown below, consists of the import substrate, the ATOM and the TIM complexes " (lines 218-219).

5- Parasites expressing LDH-DHFR-HA were used for IP experiments. However, parasites expressing LDH-DHFR-myc were used for IF. Why using different tagged- LDH-DHFR in different experiments?

LDH-DHFR-myc and LDH-DHFR-HA behave the same. At the time the IF analysis was done only LDH-DHFR-myc was available.

6- To identify novel trypanosomal TIM complex subunits, the authors performed SILAC-based quantitative IP MS analysis. For this purpose induced and uninduced parasites were mixed and then samples for IP were prepared. My concern is on how the proteins from the uninduced parasites (lacking the tagged bait protein) will bind to the affinity purification resin in order to be purified and

detected by MS.

We are not sure we understood the question. - The big advantage of SILAC-IP combined with MS analysis is that a mixed extract from uninduced and induced parasites can be applied to the matrix bound anti-HA antibody in a single tube. Due to the labeling of the control and the experimental samples with different amino acid isotopes the origin of each of the recovered peptides in the mixture can be traced to the original culture by looking at its molecular weight. This allows very accurate relative quantification of the abundance of each detected protein in the control and test samples. Thus, specifically bound proteins, copurifying with the tagged variant, will show - depending on the labeling protocol - an enrichment of either heavy or light peptides, whereas for experimental contaminants binding in absence and presence of the tagged version the same amounts of heavy and light peptides will be recovered.

About the statistical analysis, even though I am not a specialist, I think it was done correctly throughout the paper. The statistical tests are appropriated as well as the treatment of uncertainties.

Reviewer #1 (Remarks to the Author)

The authors satisfyingly addressed all points raised on the initial submission.

Reviewer #2 (Remarks to the Author)

Harsman et al made a great effort in the response letter to address the concerns raised by the three reviewers. This certainly also led to clarification of some points in the revised version of the manuscript. However, in my opinion, newly included Figure 6 brings into considerable doubt the whole scenario of a single TIM complex in *T. brucei*. Namely, new Figure 6 now shows that TbTim17, TbTim9 and TbTim42, a component identified by the authors as a new subunit of the TIM complex, can be efficiently co-purified with an arrested intermediate of a MCP12 variant, a member of the mitochondrial carrier protein family. In contrast, TimRhom I, another newly identified subunit of the TIM complex for which the authors even postulate that it "may contribute to pore formation", was not recovered at all in the same IP. In contrast, TimRhom I was specifically enriched when presequence-containing protein intermediate was analyzed. The authors finish the Results section by stating that TimRhomI "may be specifically required for the presequence pathway". I understand this sentence in the way that the authors suggest the presence of two TIM translocases and I find it hard to reconcile this statement with the abstract and the discussion which still claim the presence of "a single TIM complex which mediates import of both presequence-containing proteins as well as MCPs". In my opinion, the manuscript is unfortunately still in a too preliminary stage for publication. The authors should first clarify the composition of the MCP-translocase and carefully analyze which subunits are needed for import of which preproteins.

Minor points

1. The Results section still states that TbTim17 can be identified in SILAC-IP of TbTim42 even though the authors confirmed in the response letter that this is not the case. In the revised version the authors modified only the Figure legend which now reads „the three bait proteins ... reciprocally interact ... by SILAC-IPs or by immunoblot...“. I am not sure how the readers should be able to judge this statement if Figure 1 (for review only) is not included in the manuscript.
2. I think that Figure 2 (for review only), if reproducible and additionally carefully controlled with a parallel IP using a preimmune serum, would represent a strong argument for a specific interaction of TimRhom I and TbTim17. I am personally not disturbed by the distortions of the gel and think that one can nicely see in the Figure a depletion of TbTim17 from the unbound fraction and its enrichment in the eluate of the IP with anti-TimRhom I antibodies. Though % of load and unbound fractions, compared to eluate, were not given in the corresponding Figure Legend, it looks as if enrichments of TbTim17 and TimRhomI were comparable. This is, to me, a strong support for the SILAC-IP data. This experiment additionally addresses my major concern with SILAC IP data, which, I believe, was also shared by the Reviewer 3, and that is that the SILAC IP, in its currently presented form, only analyses eluate fractions. It is thus impossible to judge enrichments compared to input but only compared to eluate in mock IP, in which anyway no protein should bind. For membrane proteins, the problem of nonspecific binding usually does not relate to their nonspecific binding to the beads directly but rather to their nonspecific presence in detergent micelles. The latter results in binding of membrane proteins to the beads via specific binding of the tagged protein present in the same micelle and thus binding of unrelated membrane proteins appears equally specific. In such cases, the real specificity of the binding can only be assessed by analyzing their enrichment over input – proteins specifically interacting with tagged proteins are enriched to a comparable extent as tagged proteins themselves whereas proteins binding only because they happened to be in the same micelle usually show negligible enrichments, if any.
3. concerning the answer to the suggestion to perform an antibody shift assay coupled to BNPAGE, the argument that the experiment can not be done as the affinity purified antibodies are not sufficiently concentrated sounds very weak to me - antibodies can usually be easily concentrated. Whether one will see something or not is another issue but it is certainly difficult to say anything in this direction before the experiment has been tried.

Reviewer #3 (Remarks to the Author)

The authors have now addressed all the previously raised concerns (mine and from other reviewers). The manuscript has been improved in the revision process. I strongly support the publication of this paper in Nature Communications.

Minor point

Figure 3b: the TimRhomII panel lacks the antibody (myc)

Reviewers' comments:

Reviewer #1 (Remarks to the Author):

The authors satisfyingly addressed all points raised on the initial submission.

Reviewer #2 (Remarks to the Author):

*Harsman et al made a great effort in the response letter to address the concerns raised by the three reviewers. This certainly also led to clarification of some points in the revised version of the manuscript. However, in my opinion, newly included Figure 6 brings into considerable doubt the whole scenario of a single TIM complex in *T. brucei*. Namely, new Figure 6 now shows that TbTim17, TbTim9 and TbTim42, a component identified by the authors as a new subunit of the TIM complex, can be efficiently co-purified with an arrested intermediate of a MCP12 variant, a member of the mitochondrial carrier protein family. In contrast, TimRhom I, another newly identified subunit of the TIM complex for which the authors even postulate that it “may contribute to pore formation”, was not recovered at all in the same IP. In contrast, TimRhom I was specifically enriched when presequence-containing protein intermediate was analyzed. The authors finish the Results section by stating that TimRhomI “may be specifically required for the presequence pathway”. I understand this sentence in the way that the authors suggest the presence of two TIM translocases and I find it hard to reconcile this statement with the abstract and the discussion which still claim the presence of “a single TIM complex which mediates import of both presequence-containing proteins as well as MCPs”. In my opinion, the manuscript is unfortunately still in a too preliminary stage for publication. The authors should first clarify the composition of the MCP-translocase and carefully analyze which subunits are needed for import of which preproteins.*

We have now clarified the composition of the carrier translocase (new Fig. 6e). In a variation of the experiment done for the presequence translocase (Fig. 5d), we have used the truncated carrier that is stuck in the import pathway and combined it with a SILAC-IP MS approach. Using a cutoff of five-fold enrichment this experiment identifies 11 proteins as subunits of the active carrier translocase, 8 of them were also present in the active presequence translocase. This indicates a large overlap in the composition of the active presequence and the active carrier translocases. However, as expected based on previous results TimRhom I and TimRhom II appear to be selectively associated with the active presequence translocase.

We concede that after the first revisions and the inclusion of many new experiments, the previously submitted revised manuscript contained ambiguous statements regarding the presence of a single or two distinct trypanosomal mitochondrial inner membrane translocases. We have now carefully edited the re-revised manuscript, slightly modified the title and re-written the abstract to clarify these points.

With the new results it becomes clear that there is a large overlap in the composition of the two active inner membrane translocases, but that there is also compositional variation as exemplified by the two rhomboid-like proteins that are specific for the presequence translocase.

Minor points

1. The Results section still states that TbTim17 can be identified in SILAC-IP of TbTim42 even though the authors confirmed in the response letter that this is not the case. In the revised version the authors modified only the Figure legend which now reads „the three bait proteins ... reciprocally interact ... by SILAC-IPs or by immunoblot...“. I am not sure how the readers should be able to judge this statement if Figure 1 (for review only) is not included in the manuscript.

We have now included the Figure in question as Supplementary Fig. 1b in the re-revised manuscript and commented on it in the main text as well.

2. I think that Figure 2 (for review only), if reproducible and additionally carefully controlled with a parallel IP using a preimmune serum, would represent a strong argument for a specific interaction of TimRhom I and TbTim17. I am personally not disturbed by the distortions of the gel and think that one can nicely see in the Figure a depletion of TbTim17 from the unbound fraction and its enrichment in the eluate of the IP with anti-TimRhom I antibodies. Though % of load and unbound fractions, compared to eluate, were not given in the corresponding Figure Legend, it looks as if enrichments of TbTim17 and TimRhomI were comparable. This is, to me, a strong support for the SILAC-IP data. This experiment additionally addresses my major concern with SILAC IP data, which, I believe, was also shared by the Reviewer 3, and that is that the SILAC IP, in its currently presented form, only analyses eluate fractions. It is thus impossible to judge enrichments compared to input but only compared to eluate in mock IP, in which anyway no protein should bind. For membrane proteins, the problem of nonspecific binding usually does not relate to their nonspecific binding to the beads directly but rather to their nonspecific presence in detergent micelles. The latter results in binding of membrane proteins to the beads via specific binding of the tagged protein present in the same micelle and thus binding of unrelated membrane proteins appears equally specific. In such cases, the real specificity of the binding can only be assessed by analyzing their enrichment over input – proteins specifically interacting with tagged proteins are enriched to a comparable extent as tagged proteins themselves whereas proteins binding only because they happened to be in the same micelle usually show negligible enrichments, if any.

The experiment shown in "Figure 2 (for review only)" is reproducible. It has now been included in the revised manuscript as Supplementary Fig. 1c. The control with the pre-immune serum is also shown as requested.

3. concerning the answer to the suggestion to perform an antibody shift assay coupled to BNPAGE, the argument that the experiment can not be done as the affinity purified antibodies are not sufficiently concentrated sounds very weak to me - antibodies can usually be easily concentrated. Whether one will see something or not is another issue but it is certainly difficult to say anything in this direction before the experiment has been tried.

It is true that among the numerous experiments suggested we decided not to do the antibody shift experiment. Based on our previous experience we judged that, considering the technical difficulty and the large volume of antiserum we would need for the experiment, it would not be worth doing. Also, the best possible result would in our opinion simply confirm that TimRhom II is present in the presequence translocase complex. However, there is already ample evidence for this as the protein is recovered in pull down experiments using four different baits.

Reviewer #3 (Remarks to the Author):

The authors have now addressed all the previously raised concerns (mine and from other reviewers). The manuscript has been improved in the revision process. I strongly support the publication of this paper in Nature Communications.

Minor point

Figure 3b: the TimRhomII panel lacks the antibody (myc)

We have carefully checked Figure 3b and think it is correct.

Reviewer #2 (Remarks to the Author):

In the re-revised version of the manuscript, the authors added new experiments and modified the text. These changes clarified my major concerns and, in my opinion, significantly improved the manuscript. I am thus happy to support its publication.

Response to REVIEWERS' COMMENTS:

Reviewer #2 (Remarks to the Author):

In the re-revised version of the manuscript, the authors added new experiments and modified the text. These changes clarified my major concerns and, in my opinion, significantly improved the manuscript. I am thus happy to support its publication.

We thank the reviewer for supporting publication of our improved manuscript.